# Predicting the Energy Landscape of Stochastic Dynamical System via Physics-informed Self-supervised Learning

**Ruikun Li**
Shenzhen International Graduate School
Tsinghua University
lirk612@gmail.com

**Huandong Wang**[*]
Department of Electronic Engineering
BNRist, Tsinghua University
wanghuandong@tsinghua.edu.cn

**Qingmin Liao**
Shenzhen International Graduate School
Tsinghua University
liaoqm@tsinghua.edu.cn

**Yong Li**
Department of Electronic Engineering
BNRist, Tsinghua University
liyong07@tsinghua.edu.cn

## Abstract

Energy landscapes play a crucial role in shaping dynamics of many real-world complex systems. System evolution is often modeled as particles moving on a landscape under the combined effect of energy-driven drift and noise-induced diffusion, where the energy governs the long-term motion of the particles. Estimating the energy landscape of a system has been a longstanding interdisciplinary challenge, hindered by the high operational costs or the difficulty of obtaining supervisory signals. Therefore, the question of how to infer the energy landscape in the absence of true energy values is critical. In this paper, we propose a physics-informed self-supervised learning method to learn the energy landscape from the evolution trajectories of the system. It first maps the system state from the observation space to a discrete landscape space by an adaptive codebook, and then explicitly integrates energy into the graph neural Fokker-Planck equation, enabling the joint learning of energy estimation and evolution prediction. Experimental results across interdisciplinary systems demonstrate that our estimated energy has a correlation coefficient above 0.9 with the ground truth, and evolution prediction accuracy exceeds the baseline by an average of 17.65%. The code is available at github.com/tsinghua-fib-lab/PESLA.

## 1 Introduction

Energy landscapes are inherent in many stochastic dynamical systems in nature, such as the potential energy surface of protein conformations (Norn et al., 2021), the fitness landscape of species evolution (Papkou et al., 2023; Poelwijk et al., 2007), and the fractal energy landscapes of soft glassy materials. The evolution of these systems can be modeled as particles moving on the landscape under the combined effect of energy-driven drift and noise-induced diffusion. The structure of the energy landscape governs the long-term motion of particles, forming the deterministic aspect of the dynamics, while inherent random noise disrupts the movement along the energy gradient, driving exploration across energy barriers (Blount et al., 2018; Kryazhimskiy et al., 2014). When multiple low-energy regions exist in the landscape, the combined effect of the energy gradient and noise induces high-frequency movement within individual regions and low-frequency transitions between different regions (Lin et al., 2024). In this context, energy landscapes have been applied to guide the generation of stable molecular structures (Noé et al., 2019) and direct the evolution of proteins (Packer & Liu, 2015; Greenbury et al., 2022), and more recently, they have been incorporated as physical knowledge into deep learning for predicting system evolution (Guan et al., 2024; Wang et al., 2024b; Ding et al., 2024).

---
[*]Corresponding author

Due to its fundamental role in governing the system dynamics, estimating the energy landscape of dynamical systems has become an essential research problem across various disciplines. Couce et al. (2024) cultivate 50,000 generations of bacteria to measure the fitness effects of mutations, while Sarkisyan et al. (2016) measure tens of thousands of luminescent protein genotypic sequences to construct the functional landscape. These manual experimental approaches are not only costly but also heavily reliant on expert knowledge. With the success of deep learning in numerous disciplines (Jumper et al., 2021; Han et al., 2023; Wang et al., 2023; Chen et al., 2024), several deep learning models have been proposed to estimate energy or equivalent quantities based on molecular spatial structures (Zhang et al., 2018), species genotypes (Tonner et al., 2022), or population compositions (Skwara et al., 2023). These methods still require high-cost annotations to provide supervisory signals for energy, which limits their practicality. In real-world scenarios, it is typically more accessible to obtain abundant low-cost evolutionary trajectories of the system, which inherently embeds information about energy-driven drift (Weinstein et al., 2022). Therefore, an important research question arises: can we estimate the energy landscape only based on the system's evolution trajectories in a data-driven manner?

However, estimating the energy landscape from evolutionary trajectories remains a complicated problem with the following challenges. First, observable evolutionary trajectories typically cover only a limited portion of the vast state space. For instance, there are approximately $10^{11}$ potential triple mutants of a typical protein, while available high-throughput measurement techniques can only handle around $10^4$ to $10^7$ distinct genotypes, covering just a small fraction of the mutational space surrounding the natural sequence (Tonner et al., 2022). Second, distilling energy information from evolutionary trajectories requires building a model incorporating the energy landscape and the distribution of trajectory data, thereby establishing connections between them. Classical Markov state models (Noé et al., 2019) establish this connection by strictly assuming that sampled data follow a Boltzmann distribution derived from the energy, which unrealistically demands that trajectories are fully sampled from a thermodynamic equilibrium state. In contrast, existing self-supervised learning methods (Kamyshanska & Memisevic, 2014) treat neural networks as black-box models to fit data distributions, completely disregarding the guidance of physical knowledge in terms of energy and system evaluation. Currently, there is still no effective model that organically integrates AI techniques and physical knowledge for energy estimation without supervisory signals.

In this paper, we propose a Physics-informed Energy Self-supervised Landscape Analysis (PESLA) method to estimate the energy landscape from historical evolution trajectories in a self-supervised manner. PESLA maps the system state from the observed space to a discretized latent space via vector quantization techniques (Van Den Oord et al., 2017). Through adaptively learning a codebook to partition the vast state space, our model concentrates on the essential shapes of the energy landscape in discrete domains, thus disregarding the negligible information of the energy landscape and overcoming the challenge posed by limited observations. Then, PESLA utilizes the self-supervision signal from the prediction error of the system state to guide energy estimation. In this process, a graph neural ODE inspired by the Fokker-Planck equation is utilized to model the time evolution of probability distributions across different discretized states, and a physics-inspired regularization constraint is employed to integrate the prior knowledge of Boltzmann distribution of long-term dynamics (Sato & Nakagawa, 2014), without relying on the assumption of thermodynamic equilibrium sampling. These physics-inspired architectures serve as the bridge to distill information of the energy landscape from the system dynamics, thereby enabling the self-supervised learning of the system's energy landscape.

Our contribution can be summarized as follows:

- We develop a novel framework to estimate the energy landscape of the system only utilizing the self-supervision signal from predicting the system state, where the physics-information architecture of graph neural Fokker-Planck equation and physics-inspired regularization serves as the bridge to distill information of energy landscape from the system dynamics.
- We develop a discrete encoding method of the system state to coarsen the continuous energy landscape using a codebook obtained through vector quantization techniques. It allows our model to concentrate on the essential shapes of the energy landscape, effectively disregarding its negligible information and enhancing the sample efficiency of limited observational data.
- Experimental results across interdisciplinary systems demonstrate that PESLA reliably estimates system energy with absolute correlation coefficients above 0.9 and achieves 17.65% higher evolution prediction accuracy compared to state-of-the-art baselines.

## 2 BACKGROUND AND PROBLEM SETUP

Let us consider a stochastic dynamical process which can be described by the following differential equation:

$$ds_t = f(s_t)dt + \sigma(s_t, t)dW(t), \tag{1}$$

$$x_t = g(s_t). \tag{2}$$

Specifically, it represents a system with latent state variable $s_t \in \mathcal{S}$ whose evolution is driven by a deterministic drag force $f(s_t)$ and a random force described by white noise $\sigma(s_t, t)dW(t)$. While the state variable $s_t$ is hidden and cannot be observed directly, the observable measurement $x_t \in \mathcal{X}$ of the system is derived through a transformation $g : \mathcal{S} \to \mathcal{X}$, which can be either linear or nonlinear and can even represent a mapping from continuous space to discrete space, thereby describing systems with discrete observable metrics, such as ecological evolution.

More specifically, we focus on systems where the force $f(s_t)$ is conservative. This implies the existence of an energy function $E(s_t)$, also referred to as the energy landscape, such that $f(s_t) = -\nabla E(s_t)$. Then, the dynamic equation 1 can be be rewritten as:

$$ds_t = -\nabla E(s_t)dt + \sigma(s_t, t)dW(t), \tag{3}$$

The energy landscapes measure the thermodynamic stability of a given state. Low-energy regions induce a drift that draws the system state into them with greater probability and duration, manifesting thermodynamically as the Boltzmann distribution, $p \propto e^{-E(s)/kT}$, where $k$ is Boltzmann constant and $T$ represents temperature. For evolution starting from any initial state distribution, the system's long-term dynamics will eventually drift toward the Boltzmann distribution defined by the energy landscape. Examples of such energy landscapes in different disciplines include fitness landscapes in ecology Papkou et al. (2023), potential energy in molecular dynamics Chmiela et al. (2017), and free energy in glassy materials Charbonneau et al. (2014).

**Learning problem** In this paper, our primary objective is to estimate the energy landscape of a stochastic dynamical system based on its evolution trajectories, without the true energy as a supervisory signal. More formally, the input of this learning problem is a set of the $N$-step evolution trajectory $X_N = \{x_{t_i}\}_{i=0}^{N-1}$ of the stochastic dynamical system in the $D$-dimensional observation space $\mathcal{X}$. Then, for an arbitrary observable state $x$, the objective of this learning problem is twofold: (1) building a transformation $\mathcal{E}$ to map the observable measurement to a latent feature $c = \mathcal{E}(x)$ that determines the energy of the system; (2) estimating the energy $\hat{E}(\mathcal{E}(x))$ as an approximation of the true energy $E(g^{-1}(x))$. Since the true energy $E(g^{-1}(x))$ is unavailable as a supervisory signal in the learning process, the estimated energy $\hat{E}(\mathcal{E}(x))$ is only required to be a linear transformation of the true energy.

## 3 METHOD

In this section, we introduce a Physics-informed Energy Self-supervised Landscape Analysis (PESLA) method, which learns to predict the energy landscape through a self-supervised evolution prediction task, as shown in Figure 1. First, we develop an adaptive codebook learning module to instantiate the mapping $\mathcal{E}$ from the observed space to the energy landscape. This approach integrates concepts from reduced-order modeling of complex systems to mitigate uncertainties caused by limited sample coverage. Next, we explicitly incorporate the energy function into a graph neural Fokker-Planck equation to model the system's evolution on the energy landscape. Additionally, we introduce physics-inspired regularization constraints into the optimization objective to eliminate the assumption of thermodynamic equilibrium sampling.

### 3.1 ADAPTIVE CODEBOOK LEARNING

Constructing the energy landscape involves learning the transformation $\mathcal{E}$ from the observed space $\mathcal{X}$ to the latent space $\mathcal{S}$ where the energy landscape resides. Previous studies have shown that, despite the high dimensionality of the state space, the long-term dynamics of systems unfold on a very low-dimensional manifold in the form of reduced-order model (Vlachas et al., 2022; Thibeault et al., 2024; Li et al., 2024). This suggests that the energy landscape, which shapes the system's

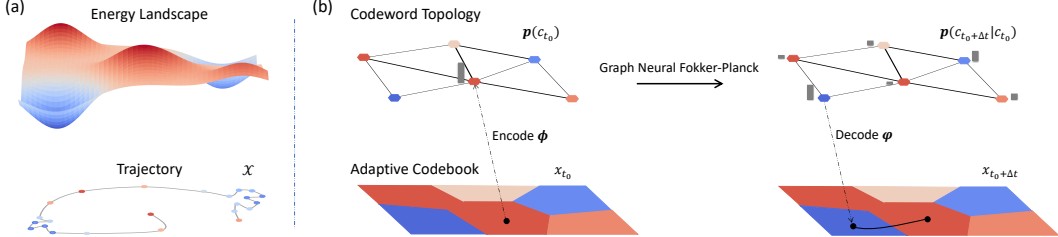

Figure 1: Framework of PESLA. (a) The energy landscape with evolution trajectories; (b) Partitioning the state space with an adaptive codebook to form the codewords with a graph topology and modeling the time evolution of probability across neighboring regions by graph neural Fokker-Planck equation.

long-term evolution, has inherently low dimensionality. Similar phenomena have been observed in natural language processing and image representation, where a set of discrete codewords is sufficient to capture the essential representation of the original data (Van Den Oord et al., 2017; Razavi et al., 2019). Therefore, modeling the energy landscape as a discrete reduced-order model in the latent space $\mathcal{S}$ offers a promising approach to addressing the challenge of the vast state space (Noé et al., 2009).

To implement such reduced-order approach and identify the energy landscape in the latent space $\mathcal{S}$, we enhance the autoencoder with a learnable codebook $C = \{c_i \in \mathbb{R}^d \mid i = 1, 2, ..., K\}$ to discretize the latent space of the encoded data. The transformations $\mathcal{E}$ and $g$ between the observed space and the latent space are parameterized by $\Xi$ and $\Omega$, respectively. Specifically, after a sample $x$ is encoded into a latent vector $s$, it is mapped to the most similar codeword $c_i$, which then serves as the input to the probabilistic decoder $\Omega$ for reconstructing $x$. This k-nearest neighbor (KNN) style discrete aggregation partitions the latent space into multiple local regions (as shown in Figure 1b), each uniquely represented by the energy of a codeword, thereby forming the low-dimensional landscape space. The encoding function $\Xi$ maps the original space to the landscape space, capturing semantic features to ensure similar states fall into the same codeword region, thereby reducing reconstruction error. We emphasize that this design allows for optimal scaling of the state space partitioning from the limited coverage of observed data, rather than simple equidistant grid binning, as shown in Figure 3a (center). This adaptive scaling ensures the maximal utilization of codewords, enhancing the robustness to the preset number of codewords. Through the adaptive codebook encoding, the observed trajectories are mapped onto the energy landscape in the form of codewords, i.e., $\{c_{t_i}\}_{i=0}^{N-1}$.

## 3.2 GRAPH NEURAL FOKKER-PLANCK EQUATION

In the latent space $\mathcal{S}$, the time evolution of the system state is influenced by the combined effect of energy-driven drift and diffusion caused by inherent random noise, theoretically modeled by the Fokker-Planck equation (Risken, 1996). On the discretized energy landscape, we extend the traditional Fokker-Planck equation into a graph neural differential equation, enabling joint learning of energy estimation and evolution prediction.

We construct the codeword topology $A = (a_{ij})_{K \times K}$ based on the adjacency relationships of the codeword regions (as shown in Figure 1) and estimate the energy of each codeword as $E(c_i)$ as the energy landscape $\mathcal{G} = \{A, C, E(*)\}$ of system evolution. At this point, we have projected the original observed trajectory onto a low-dimensional energy landscape, obtaining the transition trajectory of the system state on the codeword topology. Predicting the temporal evolution of the system means modeling the time-dependent evolution of the probability distribution over the codewords. The effects of energy and noise on this evolution are modeled by the Graph Fokker-Planck equation Chow et al. (2012) as:

$$\frac{dp_i}{dt} = \sum_{j \in N(i), E_{ji} > 0} ((E_{ji} + \beta \log \frac{p_j}{p_i})p_j + \sum_{j \in N(i), E_{ji} < 0} ((E_{ji} + \beta \log \frac{p_j}{p_i})p_i + \sum_{j \in N(i), E_{ji} = 0} \beta(p_j - p_i),$$

(4)

where $p_i$ denotes the probability of node $i$ and $E_{ji} = E_j - E_i$. $\beta$ is a positive constant which governs the noise strength. Denoting $\mathbf{p}(c_{t_0})$ as $K$-dimensional probability distribution at time $t_0$,

one can naively obtain the conditional probability distribution $\mathbf{p}(c_{t_0+\Delta t}|c_{t_0})$ after a diffusion time of $\delta t$ by performing a time integration of the Fokker-Planck equation on the initial condition $\mathbf{p}(c_{t_0})$.

However, considering that the evolution of a node's state often depends on its neighbors, projecting the one-dimensional probability vector into a higher-dimensional space with stronger representational capacity helps capture this rich relational structure. We employ a graph convolutional neural networks (GCN) based probability encoder, $\mathbf{H}(t_0) = \Phi(\mathbf{p}(c_{t_0}))$, introducing neighborhood information through positional encoding (Chamberlain et al., 2021b; Yuan et al., 2024a) to lift the one-dimensional probability vector into a high-dimensional representation. Thus, the time evolution of conditional state probabilities $\mathbf{p}(x_{t_0+\Delta t}|x_{t_0})$ on the energy landscape $\mathcal{G}$ is modeled as a graph neural diffusion process (Chamberlain et al., 2021a; Yuan et al., 2024b), formalized as

$$
\begin{aligned}
\mathbf{H}(t_0 + \Delta t) &= \mathbf{H}(t_0) + \int_{t_0}^{t_0+\Delta t} \frac{\partial \mathbf{H}(t)}{\partial t} dt, \\
\mathbf{p}(c_{t_0+\Delta t}|c_{t_0}) &= \Psi(\mathbf{H}(t_0 + \Delta t)).
\end{aligned}
\tag{5}
$$

We extend Chow et al. (2012)'s theory by designing a graph neural Fokker-Planck equation to explicitly model state diffusion driven by energy differences between neighboring codewords as

$$
\frac{\partial}{\partial t}\mathbf{H}_{c_i} = \sum_j \mathbf{W}_{ij}[E_{ji} + \beta_\xi(\log \mathbf{H}_{c_j} - \log \mathbf{H}_{c_i})] \circ [\sigma(kE_{ji})\mathbf{H}_{c_j} + (1 - \sigma(kE_{ji}))\mathbf{H}_{c_i}],
\tag{6}
$$

where $\mathbf{W}_{ij}$ is calculated by neighborhood attention. The learnable coefficient $\beta_\xi$ represents the strength of noise acting between neighboring codewords, while $k$ is the scaling factor for the sigmoid activation function. The ablation study can be found in Appendix E, where we demonstrate that modeling in the encoded probability space performs significantly better than directly modeling the probability vector.

### 3.3 TRAINING

The trainable parameters include the encoder $\Xi$, decoder $\Omega$, codebook $C$, probability encoder $\Phi$, probability decoder $\Psi$, neighborhood attention weights $W$, coefficient vector $\beta_\xi$, and energy function $E(*)$. The detailed model architecture is provided in Appendix A.2. In the following, we introduce the training procedure for the model.

Adaptive codebook learning and evolution prediction form a joint learning task. The optimization objective for the former is to minimize the negative log-likelihood of the reconstructed distribution, i.e., $L_{reconstruct} = -\log \mathbf{q}_{\Xi,\Omega,C}(x)$. In our experiments, we use a Gaussian prior distribution decoder with negative log-likelihood loss for continuous systems, and cross-entropy loss for discrete systems. Additionally, the loss function $L_{vq}$ for updating codeword is consistent with the one proposed by Van Den Oord et al. (2017). Similarly, we minimize the negative log-likelihood in both the latent space and the landscape space for the evolution prediction task. In the latent space, we minimize the L2 error $L_{latent} = ||\Phi(\mathbf{p}(c_{t+\Delta t})) - \Psi(H(t + \Delta t))||$, while in the landscape space, we use cross-entropy $L_{code} = -\mathbf{p}(c_{t+\Delta t})\log \mathbf{q}(c_{t+\Delta t})$.

With the mapping of adaptive codebook, we can estimate the distribution $\mathbf{p}(c_i)$ of observed samples within the landscape space and employ the corresponding negative log-probability as reference energies to guide energy estimation. However, this approach fails when evolution trajectories are not sampled from a thermodynamic equilibrium state. Proven by statistical mechanics, the state probability distribution evolving in the form of Fokker-Planck equation converges to the Boltzmann distribution. Although we cannot expect all sample data to be drawn from a thermodynamic equilibrium state, the long-term evolution of states will eventually converge to the Boltzmann distribution. This suggests incorporating a regularization term into the long-term prediction task, expressed as the KL divergence between the empirical distribution $\mathbf{p}$ and the Boltzmann distribution $\mathbf{q}$, i.e., $L_{phy} = D_{\mathrm{KL}}(\mathbf{p}\|\mathbf{q}) = \sum_{i=0}^{K} \mathbf{p}(c_i)\log\left(\frac{\mathbf{p}(c_i)}{\mathbf{q}(c_i)}\right)$. Overall, we conduct the training process by optimizing the aforementioned objectives $L = L_{reconstruct} + L_{vq} + L_{latent} + L_{code} + L_{phy}$. Detailed training strategies are provided in Section 4.1 and ablation studies can be found in Appendix E.

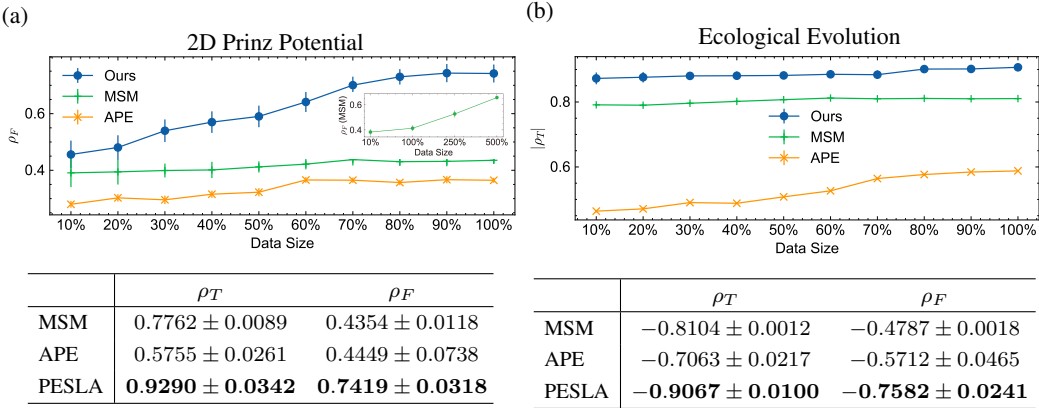

Figure 2: Visualization of the results on the energy estimation. (a): full-space energy correlation $\rho_F$ as a function of data size (top), and comparison across different methods (bottom); (b): trajectory energy correlation $\rho_T$ as a function of data size (top), and comparison across different methods (bottom).

## 4 EXPERIMENTS

We conduct experiments on three classic dynamical systems from different disciplines to evaluate the accuracy of PESLA in (1) energy estimation and (2) evolution prediction. For fairness, we use the same data preprocessing and apply grid search to fine-tune the learning rates and hyperparameters for all models. We perform 10 independent training and testing runs for each model to calculate the mean and standard deviation of all evaluation metrics in each experiment.

### 4.1 SETUP

**Baselines** For the energy estimation task, we employ the Markov state model (MSM) (Majewski et al., 2023) and autoencoder potential energy (APE) (Kamyshanska & Memisevic, 2014) as baselines. For the evolution prediction task, we compare PESLA with NeuralMJP (Seifner & Sánchez, 2023), T-IB (Federici et al., 2024), VAMPNets (Mardt et al., 2018), and SDE-Net (Kong et al., 2020). Details on the implementation and hyperparameter searching of these baseline algorithms can be found in Appendix A.3.

**Evaluation Metrics** We evaluate the accuracy of energy estimation from two perspectives. The trajectory energy correlation $\rho_T$ represents the Pearson correlation coefficient between the predicted and true energies for all samples along a new trajectory, assessing predictive performance within the regions covered by training data. The full-space energy correlation $\rho_F$ measures the correlation coefficient for the energy of system states uniformly across the entire state space, accounting for unseen areas during training. For the evolution prediction task, all metrics are measured from $M$ reference trajectories $X_N^{\tau M}$ unfolding from randomly initialized system states, where $\tau$ denotes the lag time of each step. All models are tasked with predicting evolution trajectories starting from these initial states, covering the same time span as the reference trajectories. We evaluate the accuracy of the predicted distributions by calculating the Jensen-Shannon divergence between the marginal ($MJS$) and transition ($TJS@\tau$) probability distributions of the predicted and reference trajectories across all states. For systems with a continuous state space, we discretize it into evenly spaced grid partitions, following previous work Federici et al. (2024); Arts et al. (2023). Further details can be found in Appendix A.5.

**Training strategy** We first train encoder $\Xi$, decoder $\Omega$ and the feature vectors of the codewords $C$ to construct the landscape topology. Then, we freeze them and train the parameters of the graph neural Fokker-Planck equation and energy function $E(*)$ on the landscape. For all models, we use the Adam optimizer, with the learning rate decaying exponentially by a factor of 0.99 each epoch.

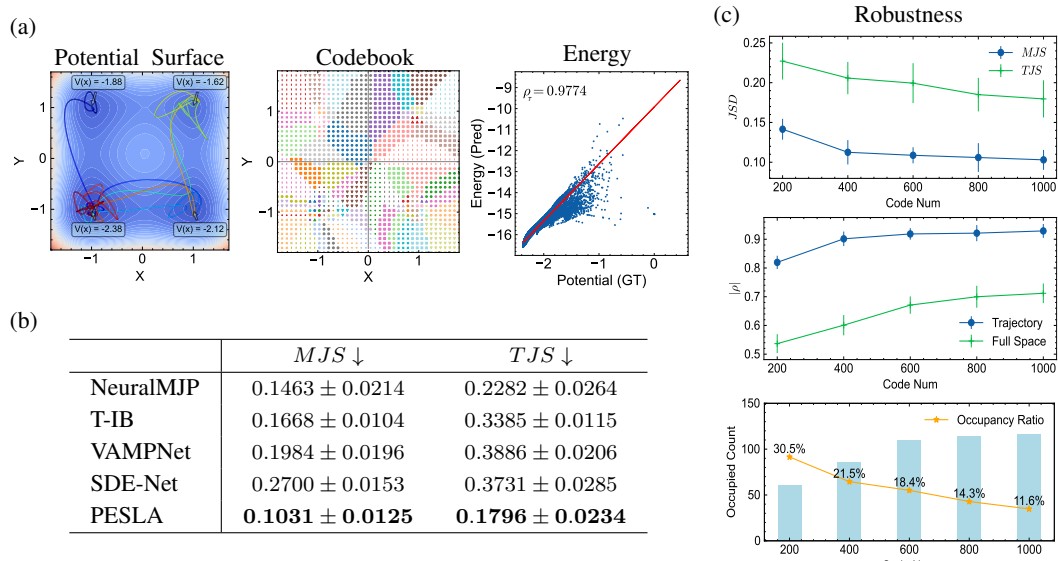

Figure 3: Visualization of the results on the 2D Prinz Potential. (a): potential surface and short sample trajectory (left), codebook distinguished by color and shape (center), and comparison between estimated energy and ground truth (right). Blue/red represents low/high values in the heatmap; (b): measures of marginal and transition JS divergence for unfolded sequences at the lag time $\tau$ of 100; (c): The impact of the preset number of codewords on evolution prediction accuracy (top), energy estimation accuracy (center), and codeword occupancy (bottom).

## 4.2 2D PRINZ POTENTIAL

We first apply PESLA to the 2D particle movement system on an asymmetric potential energy surface (Mardt et al., 2018; Federici et al., 2024). The particle displacement is governed by the stochastic differential equation as $dX_t = \nabla V(X_t)dr + \sigma dW_t$, where the potential energy function $V$, defined by $V(x) = (x_1^4 - \frac{x_1^3}{16} - 2x_1^2 + \frac{3x_1}{16}) + (x_2^4 - \frac{x_1^3}{8} - 2x_1^2 + \frac{3x_1}{8})$, consists of four interconnected low-energy regions, as shown in Figure 3a (left). A total of 10 trajectories with 100K time steps are generated and details on the generation and preprocessing can be found in Appendix A.4. The results of energy estimation and evolution prediction are presented in Figure 2a (bottom) and Figure 3b, respectively, where PESLA significantly outperforms the baseline methods in both tasks.

Figure 3a (center) visualizes the adaptive codebook learned by PESLA from historical trajectories, with different codewords distinguished by color and shape, representing their mapped regions in the original state space. The varying density of codewords at different locations directly reflects PESLA's adaptive scaling partitioning. We emphasize that the adaptive codebook captures the dynamical knowledge of the energy landscape, which is fundamentally different from the simple equidistant grid-based binning approach. At a macro level, the codebook divides the plane into an approximate 2×2 region corresponding to the four potential wells. The high energy barriers between potential wells serve as the boundaries of the four codeword regions. Low-energy wells are assigned more codewords (e.g., the bottom-left well), suggesting that the model recognizes the importance of low-energy regions as long-term dynamic attractors and allocates more *attention* to them, which aligns with the higher accuracy observed in low-energy regions shown in Figure 3a (right). We present the codebooks of multiple independent experiments in Appendix B, demonstrating that this is not a coincidental phenomenon. At a finer level, multiple codewords are assigned to the center of each potential well, while the outer areas are divided into mapped regions approximately perpendicular to the equipotential lines. This indicates that the random walk behavior induced by noise is effectively captured through the differences between codewords.

We examine the sensitivity of PESLA to data size and hyperparameters. Figure 2a (top) illustrates the impact of training data size on performance when all energy estimation algorithms are required to estimate across full-space samples. Due to the coarse-graining of adaptive codebook, PESLA

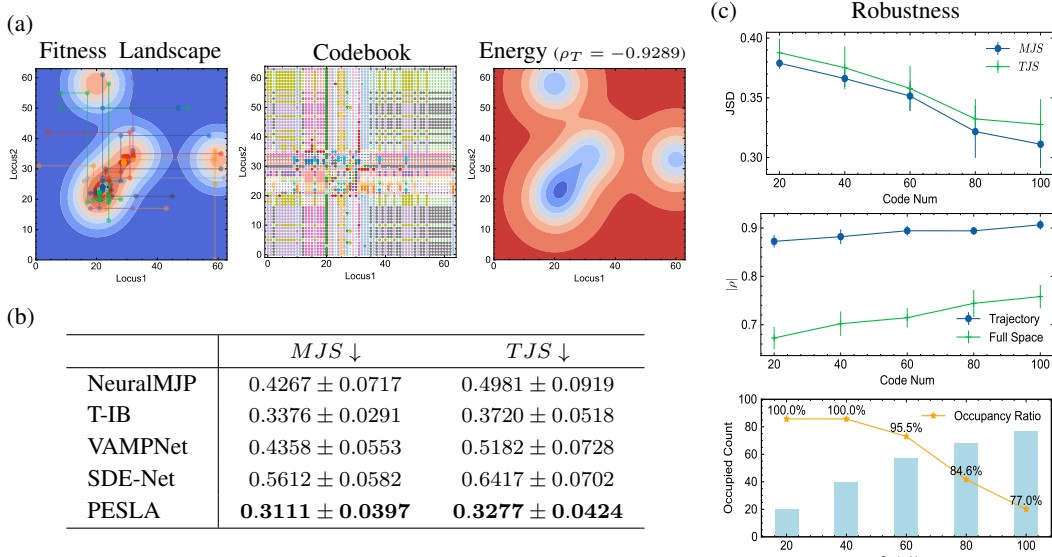

Figure 4: Visualization of the results on the Ecological Evolution. (a): fitness landscape and short sample trajectory (left), codebook distinguished by color and shape (center), and energy landscape fitted by RANSAC regression from estimated energy (right). Blue/red represents low/high values in the heatmap; (b): measures of marginal and transition JS divergence for unfolded sequences at the lag time $\tau$ of 10; (c): The impact of the preset number of codewords on evolution prediction accuracy (top), energy estimation accuracy (center), and codeword occupancy (bottom).

maintains optimal performance even with reduced data. In contrast, baseline methods are significantly limited by insufficient sample coverage in the state space. As shown in Figure 2a (top), as the data volume increases, the performance of MSM improves due to the enhanced sample coverage. Figure 3c reports the robustness of PESLA concerning the preset number of codewords. Although the number of preset codewords can be continuously increased, the actual number of occupied codewords converges automatically, and the accuracy of energy and evolution predictions reaches its peak.

## 4.3 ECOLOGICAL EVOLUTION

We examine the strong selection weak mutation system within eco-evolutionary dynamics, which is widely studied in ecology to understand the adaptive evolution of populations in specific environments (Kryazhimskiy et al., 2009; Bank et al., 2016). The fixation probability of a candidate state $j$ (new mutation) with fitness $f_j$ is governed by the Kimura formula derived from the Wright–Fisher model, given by $p_{i \to j} = \frac{1 - e^{2s_i(j)}}{1 - e^{2Ns_i(j)}}$, where $N$ represents the population size and $s_i(j) = \frac{f_j}{f_i} - 1$ is the selection coefficient. Sella & Hirsh (2005) have mathematically demonstrated that the logarithmic fitness of such evolutionary systems aligns with the energy of thermodynamic systems. We simulate 1K trajectories, each with 100 time steps, under the two-locus setting where each locus has 64 possible mutation types as our dataset. Figure 2b (bottom) and Figure 4b respectively report PESLA's superior predictive performance for fitness and system evolution.

In ecology, fitness measures the relative advantage of a genotype and is negatively correlated with energy (Sella & Hirsh, 2005). PESLA estimates the energy function of genotypes with a correlation coefficient close to -1. The predicted energy is fitted with a RANSAC regression model (see Appendix A.5) and visualized in Figure 4a (right). The distribution pattern of codewords within the codebook indicates that PESLA successfully identifies the set of genotypes with high fitness in eco-evolutionary dynamics. Moreover, since the genotype space is characterized by the Hamming distance, states in the same row or column of the codebook are more likely to be mapped to the same codeword (Figure 4a (center)). This indicates that the adaptive codebook incorporates knowledge of system dynamics rather than relying on simple equidistant grid binning.

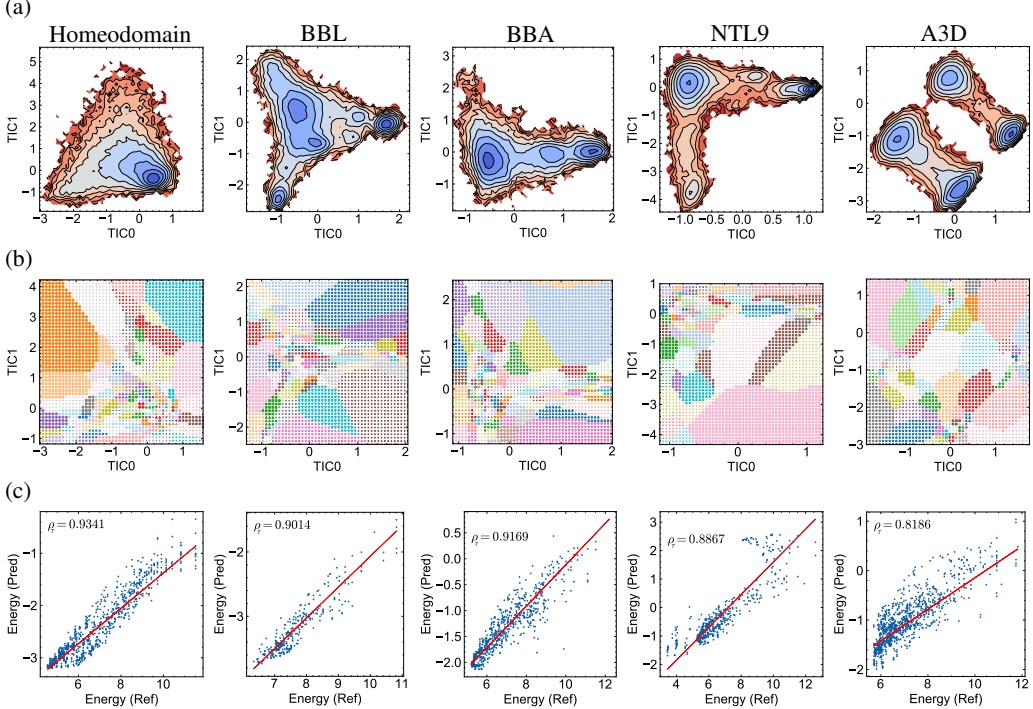

Figure 5: Visualization of the results on the Protein Folding. (a): reference energy landscapes of each protein; (b): adaptive codebooks of each protein; (c): trajectory correlation coefficients $\rho_T$ between predicted and reference energy.

We also examine the impact of data size and the preset number of codewords on PESLA in this system. For the energy estimation within the sample-covered region (measured by $\rho_T$), PESLA shows minimal sensitivity to data size, as shown in Figure 2b (top). The influence of the preset number of codewords is similar to that observed in last case, which validates PESLA's robustness.

## 4.4 PROTEIN FOLDING

We apply PESLA to the folding data of five fast-folding proteins simulated by the Anton supercomputer (Lindorff-Larsen et al., 2011). Each protein has two folding trajectories of equal length, used for model training and testing, respectively. Due to the lack of true energy, we estimate the reference energy using Time-lagged Independent Component Analysis (TICA) and the Markov State Model (MSM) based on the complete dataset (three times larger than the training data), consistent with previous studies (Majewski et al., 2023; Mardt et al., 2018). For each protein, the lag time used in TICA processing and experiments is based on the mean transition path time reported by Lindorff-Larsen et al. (2011). The reference energy distribution on the 2D principal component plane identified by TICA is shown in Figure 5a, with implementation details provided in Appendix A.4. Each protein features a varying number (1 to 4) of low-energy regions with different distributions, posing challenges for energy estimation. Figure 5b shows PESLA's partitioning of the state space for each protein on the TICA principal component plane, demonstrating that PESLA differentiates low-energy, high-energy, and unknown energy regions with varying codeword aggregation rates. This automatic scaling ensures that PESLA's energy predictions remain consistent with reference values (as shown in Figure 5c), even in challenging protein folding problems. Additionally, PESLA achieved the best performance in the evolution prediction task (see Appendix C).

## 5 RELATED WORK

### 5.1 ENERGY ESTIMATION

Estimating the energy landscape is a crucial problem across multiple disciplines. The most fundamental approach involves collecting data through modern sequencing techniques and manual experiments. Sarkisyan et al. (2016) measured tens of thousands of Aequorea victoria (avGFP) derivative genotypes to construct the local fitness landscape of green fluorescent proteins. Chen et al. (2022) analyzed the fitness of all single mutations in VIM-2 $\beta$-lactamase across a 64-fold range of ampicillin concentrations. Additionally, Wang et al. (2024a) conducted high-throughput functional genomics on Salmonella to identify gene networks related to adaptive effects. There have been many similar efforts (Starr et al., 2018). However, these manual experiments are often associated with high operational costs, making machine learning a promising solution to improve this process in a data-driven manner (Rupp et al., 2012; Han et al., 2023). Zhang et al. (2018) introduce the deep potential molecular dynamics method, using neural networks to model interatomic forces and potential energy. To mitigate overfitting issues in deep neural networks, Aghazadeh et al. (2021) apply sparse recovery algorithms from coding theory for spectral regularization. Zhang et al. (2022) employ high-speed atomic force microscopy to collect data for training a U-net model to predict the energy landscape of spatial angles on the DHR10-micaN protein. Additionally, Tonner et al. (2022) and Skwara et al. (2023) offer interpretable predictions of mutation effects and population functions through hierarchical Bayesian modeling and polynomial regression, respectively. More recently, Du et al. (2024) developed a graph neural network to model intermolecular interactions, predicting Gibbs free energy in solute-solvent interactions. Despite these efforts, these models often depend on true energy values or molecular force fields as supervisory signals. In contrast to these methods, our PESLA does not require supervisory signals for energy; instead, it learns to estimate energy through a self-supervised evolution prediction task. An additional benefit of this approach is that the predicted energy effectively enhances the accuracy of evolution prediction.

### 5.2 EVOLUTION PREDICTION

Predicting the evolution of stochastic dynamical systems is challenging due to the unknown underlying energy landscape. Vlachas et al. (2022) employ dimensionality reduction techniques to construct reduced-order models that capture essential macroscopic information, thereby simplifying the analysis of large-scale systems. To handle the challenge of modeling long-term dynamics, approaches such as learning time-invariant representations have been explored (Federici et al., 2024; Kostic et al., 2024b; Li et al., 2023). Furthermore, Kostic et al. (2022; 2024a) extend Koopman operator theory to map system states into a Hilbert space, facilitating the learning and interpretation of nonlinear dynamics. Wu et al. (2018) and Seifner & Sánchez (2023) represent stochastic dynamical processes as discrete state transitions within a Markov process framework. In contrast to existing methods, PESLA utilizes energy landscape knowledge to guide system dynamics modeling.

## 6 CONCLUSION

In this paper, we propose the PESLA method to estimate the energy landscape from historical evolution trajectories in a self-supervised manner. By integrating adaptive codebook learning and a graph neural Fokker-Planck equation, PESLA collaboratively models the energy landscape and system dynamics, even with limited observational data. We introduce physics-inspired regularization to help PESLA move beyond the reliance on thermodynamic equilibrium sampling. Experimental results across various systems demonstrate that PESLA outperforms state-of-the-art methods in both energy estimation and evolution prediction. PESLA does not require supervisory signals for energy, making it a powerful data-driven tool for understanding and predicting stochastic dynamical systems.

**Limitations and Future work** This work focuses on estimating the energy landscape of a class of energy-driven evolutionary systems. However, when a system is driven by non-conservative forces, an energy landscape does not exist, as in the case of motion in viscous fluids. Additionally, inferring energy landscapes becomes more challenging when the landscape is time-varying, such as in cases where climate change alters species fitness. Future work will need to explore additional model designs to accommodate the dynamics of time-varying landscapes, where $E(*)$ needs to adapt to $E(*, t)$.

## ACKNOWLEDGMENTS

This work was supported in part by the National Natural Science Foundation of China under U23B2030, 62171260, and 92270114. We sincerely appreciate the inspiration and valuable insights from discussions with Dr. Jiliang Hu.

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

## A    Experimental Details

We provide details on the experimental systems, baselines, evaluation metrics, and implementation to ensure clarity and reproducibility of the reported results.

### A.1    Introduction and Implementation of Baselines

Our baselines cover both the energy estimation and evolution prediction tasks. For the energy estimation task, we have:

- **Markov State Model (MSM)** is a commonly used method for estimating the relative energy of system states based on statistical probabilities. It discretizes the system using equidistant grid binning and then calculates the negative log of the frequency distribution for all states in the dataset as the reference energy. This approach is often limited by the inefficiency of Monte Carlo sampling. When the dataset fails to cover the entire sample space, some state frequencies become zero, making it impossible to infer unobserved samples from the existing data. In our experiments, we used nearest-neighbor interpolation to compute the full-space energy correlation coefficient $\rho_F$ for quantitative evaluation.

- **Autoencoder Potential Energy (APE)**. Kamyshanska & Memisevic (2014) demonstrate that an autoencoder can estimate the energy of a sample by treating the reconstruction error as a proxy for energy, where a lower reconstruction error indicates that the sample lies in a high-probability, low-energy region of the learned manifold, while a higher error corresponds to a higher energy. For an autoencoder with sigmoid activations, with weights $W$, hidden biases $b_h$, and reconstruction biases $b_r$, the energy function is given by:

$$E(x) = \sum_k \log(1 + \exp(W_k^T x + b_{h_k})) - \frac{1}{2}\|x - b_r\|^2 + \text{const},$$

  where $W_k^T x$ represents the linear combination of inputs, $b_{h_k}$ is the hidden bias term for the $k$-th hidden unit, and $b_r$ is the reconstruction bias.

For the evolution prediction task, we compare PESLA with:

- **Neural MJP**. Seifner & Sánchez (2023) introduce Neural MJP as an alternative variational inference algorithm for Markov jump processes, which relies on neural ordinary differential equations in the form of the master equation. Neural MJP predefines the number of discrete states and encodes observed states into a one-hot vector representing the discrete state distribution as the starting point for state evolution. The key difference between these predefined discrete states and PESLA's codewords is that Neural MJP does not characterize them by energy but instead relies on a black-box neural network to fit the transition probabilities. The number of preset discrete states is treated as a hyperparameter.

- **T-IB** (Federici et al., 2024) captures time-invariant representations of continuous dynamical systems using a representation learning objective derived from information bottleneck theory and models state transitions in the representation space through a conditional flow model. This efficient representation allows T-IB to filter out high-frequency fluctuations as noise and model long-term dynamics over extended time spans.

- **VAMPnet** (Mardt et al., 2018) captures the dynamics of molecular systems by directly learning a transformation from molecular configurations to a Markov state model using a deep neural network that maximizes a variational score. This end-to-end approach allows it to identify slow dynamical processes and long-timescale kinetics effectively.

- **SDE-Net** (Kong et al., 2020) explicitly models the drift and noise diffusion terms of stochastic dynamical systems by parameterizing these two mechanisms within a neural differential equation, enhancing the representational capacity of the neural network. However, SDE-Net does not treat the dynamical system as a Markov process, making it challenging to capture transition characteristics between metastable states. Despite this limitation, it serves as a benchmark for all models.

## A.2 ARCHITECTURE OF PESLA MODEL

We summarize all components of the PESLA model and the parameter shapes of each component in Table 1, where $D$ is the dimension of the observed state, $K$ is the preset number of codewords, and $r$ is the proportion of activated codewords.

Table 1: Module and layer specifications.

| Module | Layer name | Parameter shape |
|---|---|---|
| i. Adaptive Codebook Learning | | |
| Encoder $\Xi$ | Layer-FC | $[D, 64, 32]$ |
| Codebook $C$ | | $[K, 32]$ |
| Gaussian Decoder $\Omega$ | Layer-FC | $[32, 64, 64]$ |
| | Linear (mu) | $[64, D]$ |
| | Linear-Sigmoid (std) | $[64, D]$ |
| ii. Graph Neural Fokker-Planck Equation | | |
| Energy Function E(*) | Layer-FC | $[32, 64, 1]$ |
| Probability Encoder $\Phi$ | Positional Encoding | $[rK, 3]$ |
| | GCN-FC1 | $[3+1+1, 64]$ |
| | GCN-FC2 | $[64, 64]$ |
| Neighborhood Attention $W$ | Linear (q) | $[3, 64]$ |
| | Linear (k) | $[3, 64]$ |
| Coefficient Vector $\beta_\xi$ | | $[64,]$ |
| Probability Decoder $\Psi$ | GCN-FC1 | $[64, 64]$ |
| | GCN-FC2 | $[64, 1]$ |

## A.3 GRID SEARCHING FOR HYPERPARAMETERS

To ensure a fair comparison across all models, we used the same batch size, optimizer, and learning rate decay strategy during training, conducting grid search only on the learning rate and model-specific hyperparameters to achieve optimal performance. The range and targets of the hyperparameter search are detailed in Table 2.

Table 2: Hyperparameter search settings.

| Model | Learning Rate | Specific Hyperparameters | Description |
|---|---|---|---|
| NeuralMJP | $0.01 - 0.0001$ | $10 - 1000$ | The preset number of discrete states |
| T-IB | $0.01 - 0.0001$ | $0.01 - 1.0$ | Information bottleneck coefficient |
| VAMPNet | $0.01 - 0.0001$ | $10 - 1000$ | Output dimensionality of the encoder |
| SDE-Net | $0.01 - 0.0001$ | $0.01 - 1.0$ | Intensity coefficient of noise term |
| PESLA | $0.01 - 0.0001$ | $10 - 1000$ | The preset number of codewords |

## A.4 DATA GENERATION OR PREPROCESSING

Consistent with previous studies, we simulate the 2D Prinz potential and ecological evolution systems to obtain the datasets. For protein folding, we use the official data provided by the authors, as described in Table 3. For the first two systems, the training and testing sets are split in a 7:3 ratio. For the protein data, each protein has two trajectories of equal length, one used for training and the other for testing.

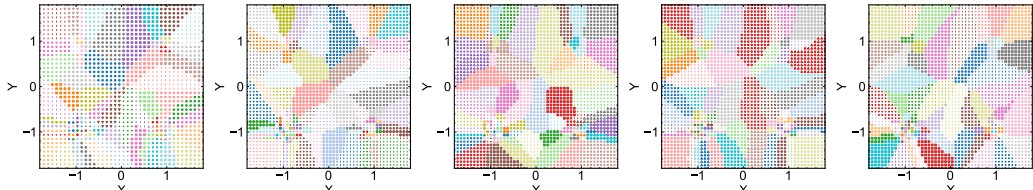

Figure 6: Visualization of the adaptive codebooks from five independent experiments of PESLA on the 2D Prinz potential.

The folding data for the five proteins includes 3D spatial coordinates of 28 to 73 alpha-carbon atoms. We performed TICA implemented by Deeptime (Hoffmann et al., 2021), extracting the linear projections of the top two principal time components for each protein as preprocessing, which were then used for estimating reference energy and testing all models.

Table 3: Protein folding dataset.

|  | Homeodomain | BBL | BBA | NTL9 | A3D |
|---|---|---|---|---|---|
| C-atom Num | 52 | 47 | 28 | 39 | 73 |
| Trajectory Length ($\mu s$) | 100 | 100 | 200 | 300 | 300 |
| Time unit ($ns$) | 10 | 10 | 10 | 10 | 10 |

### A.5 EVALUATION METRICS CALCULATION

For the evolution prediction task, we test each model using the following steps:

1. Randomly initialize $M$ initial states and predict $N$ future steps using the model to obtain $X_N^{\tau M}$;

2. Discretize each dimension into a $K \times K$ state space using a uniform grid, resulting in a finite set of discrete states;

3. Compute the marginal and transition probability distributions for each state across all $M$ trajectories;

4. Calculate the Jensen-Shannon divergence of the marginal and transition probabilities for each trajectory and take the average.

We set $K$ to 5, 8, and 8 for the 2D Prinz potential, ecological evolution, and protein folding systems, respectively.

For the trajectory energy correlation reported in Figure 4a (right), we used the Random Sample Consensus (RANSAC) regression algorithm to fit the maximum likelihood expression $E_{pred} = f(E_{true})$ of the true energy and PESLA's predicted energy. We then mapped the data from Figure 4a (left) using $f$ and visualized it in Figure 4a (right).

## B SUPPLEMENTARY EXPERIMENTAL RESULTS FOR 2D PRINZ POTENTIAL

We provide the codebooks from five independent experiments on the 2D Prinz potential in Figure 6, showing that PESLA consistently learns similar codeword distribution patterns.

## C SUPPLEMENTARY EXPERIMENTAL RESULTS FOR PROTEIN FOLDING

We provided a supplementary comparison of the predictive performance of all models using the BBA protein as an example, as shown in Table 4.

Table 4: Predictive performance of all models on the BBA protein data with a lag time of 0.7 $\mu s$ and 8×8 grid discretization. All experiments are run 10 times to obtain statistical values.

|  | $MJS \downarrow$ | $TJS \downarrow$ |
|---|---|---|
| NeuralMJP | $0.0231 \pm 0.0048$ | $0.3637 \pm 0.0051$ |
| T-IB | $0.0388 \pm 0.0053$ | $0.4327 \pm 0.0083$ |
| VAMPNet | $0.0411 \pm 0.0021$ | $0.4566 \pm 0.0228$ |
| SDE-Net | $0.0561 \pm 0.0181$ | $0.5502 \pm 0.0611$ |
| PESLA | $\mathbf{0.0207 \pm 0.0061}$ | $\mathbf{0.2468 \pm 0.0215}$ |

# D    SUPPLEMENTARY EXPERIMENTAL RESULTS FOR ADDITIONAL EXPERIMENTS

## D.1    INTERPRETABILITY

Our PESLA synergistically estimates energy and predicts trajectories to simultaneously improve the accuracy of both. Although the quality of evolution prediction directly influences the precision of energy estimation, it remains unclear how the accuracy of energy estimation, in turn, impacts evolution prediction. Here, we investigate how the correlation between the estimated energy landscape and the true energy landscape influences the evolution prediction. Specifically, we aim to clarify the degree of correlation required between the predicted energy and the true energy to ensure accurate evolution prediction.

We disable the energy prediction module of PESLA, replacing the predicted energy of each codeword with a dummy energy value. When the Pearson correlation coefficient, denoted as $\rho$, equals 1.0, the dummy energy is derived from the mean true energy values of all samples within the region of each codeword. We gradually introduce noise to the dummy energy to reduce its correlation with the true energy, as illustrated in the first row of Figure 7. Subsequently, we train PESLA under various dummy energy conditions and evaluate the prediction error. As shown in Figure 7, the prediction error progressively increases as the correlation coefficient between the dummy energy and the true energy decreases. When the correlation coefficient drops below 0.5, PESLA's predictive performance begins to lag behind the optimal baseline algorithm (NeuralMJP). This indicates that the quality of evolution prediction is directly influenced by the accuracy of energy estimation.

## D.2    CONSISTENCY

Although the degree of discretization of the state space depends on the predefined number of codewords, a robust prediction model should yield consistent energy landscapes across different settings. We evaluate the correlation between energy values predicted by PESLA under various hyperparameter settings (predefined number of codewords $K$) and random seeds. As shown in the correlation matrix in Figure 8, the energy landscapes identified by PESLA remain consistent not only across parallel experiments with different random seeds but also across different choices of hyperparameters $K$.

## D.3    NOISE ROBUSTNESS

Considering that real-world trajectory data is usually noisy and sparse, the robustness of a predictive model to noise and limited data determines its practical utility. As reported in Figure 2a of the main text, PESLA outperforms all baselines when available data is reduced. Here, we further evaluate PESLA's robustness to noisy data. Specifically, we add Gaussian noise of varying strength to the dataset, with a noise amplitude equal to the original data magnitude when the strength is set to 1.0. The results in Figure 9 indicate that PESLA remains sufficiently robust to noise until the noise strength exceeds 0.6. PESLA's robustness to noise can be attributed to its adaptive codebook learning model, which incorporates a reduced-order approach. By identifying a low-dimensional, compact representation of the original state space, PESLA inherently possesses the ability to filter out uncertainties such as noise-related errors.

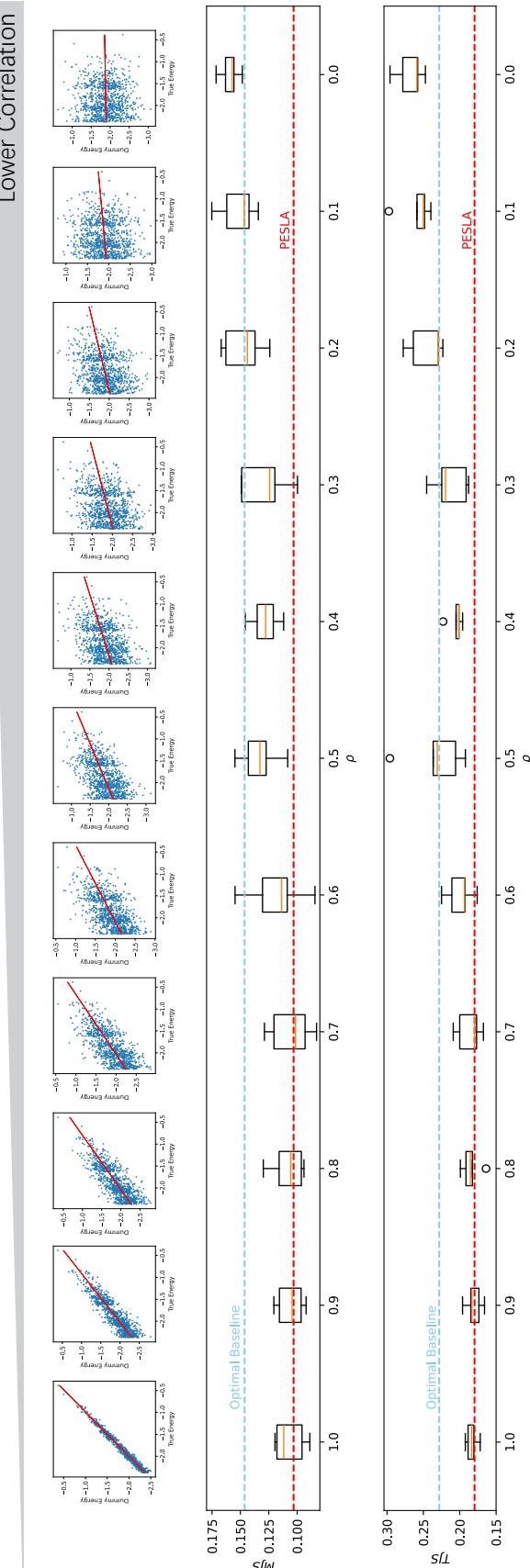

Figure 7: Evolution prediction accuracy as a function of the correlation coefficient $\rho$ between the dummy energy and the true energy on the 2D Prinz potential. The skyblue and red dashed lines are the optimal baseline and PESLA performances in the main text, respectively. All experiments are run 10 times to obtain statistical values.

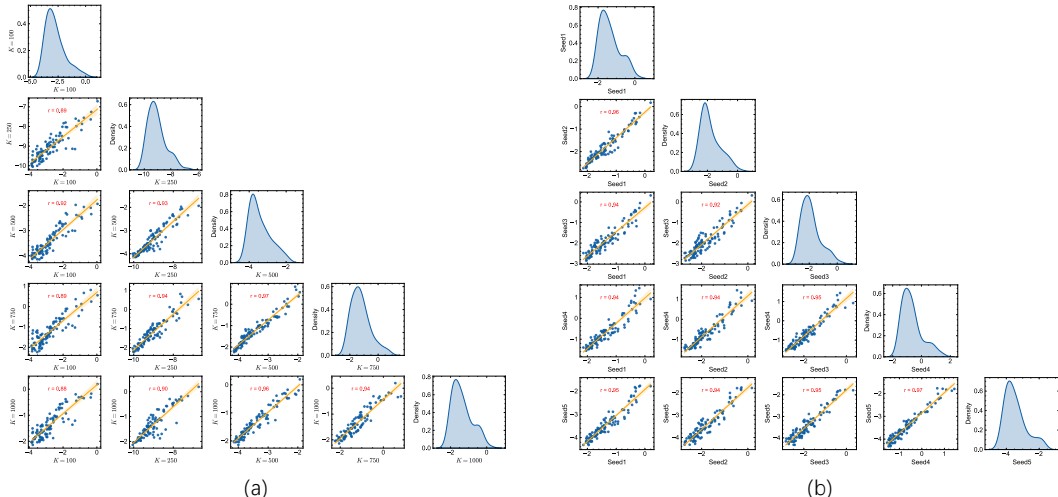

Figure 8: Correlation matrix of energy values predicted by PESLA for BBA protein at different (a) preset numbers of codewords $K$ and (b) random seeds.

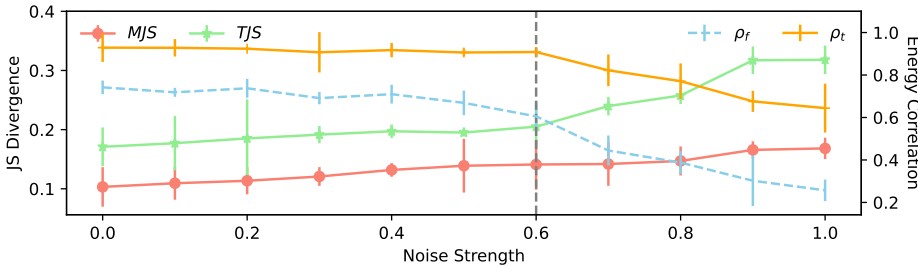

Figure 9: Energy and evolution prediction accuracy on the 2D Prinz potential as functions of noise strength. All experiments are run 10 times to obtain statistical values.

## D.4 TRANSFERABILITY

Here, we explore the transferability of PESLA. Although the energy landscapes of different evolutionary systems are inherently distinct, PESLA's modules can be partially reused in similar state spaces by learning a generalized spatial mapping mechanism. We proceed by evaluating PESLA's transferability across five different protein-folding datasets.

Due to differences in sequence length and arrangement, the folding processes of different proteins occur within their unique energy landscapes, which means that the energy function $E(*)$ and transition model need to be specifically trained for each type of protein. However, the mapping functions $\Xi$ and $\Omega$ between the observed space $\mathcal{X}$ and the latent space $\mathcal{S}$ have the potential for transferability. By learning a universal encoder $\Xi$, decoder $\Omega$ and codebook $C$, it is promising to project the structures of various proteins onto a unified latent space.

To test such transferability, we conduct cross-protein experiments for each protein. Specifically, for a given protein $i$, we train the encoder $\Xi$, decoder $\Omega$, and codebook $C$ using data from the other four proteins. We then freeze their parameters and use a single folding trajectory of protein $i$ to train the energy function $E(*)$ and the Graph Neural Fokker-Planck equation. Finally, we evaluate the accuracy of energy and evolution predictions on unseen folding trajectories of protein $i$. The results are presented in Table 5. Although the predictive performance in all cross-protein transfer experiments is lower than that achieved by training on specific proteins, the average transfer performance $\rho_t$ for energy prediction across all proteins reaches over 80% of the performance of specifically trained models. Additionally, the transfer performance of evolution prediction is significantly better

than that of the optimal baseline under the same training settings. We believe that as the number of available proteins increases, the model's transferability will show promising improvement. The validation experiments in this section demonstrate its feasibility.

Table 5: Comparison of mean $\rho_t$, MJS, and TJS metrics for encoder $\Xi$, decoder $\Omega$, and codebook $C$ trained on specific protein data versus cross-protein data for Homeodomain, BBL, BBA, NTL9, and A3D. All experiments are run 10 times to obtain mean values.

| | Homeodomain | | | BBL | | | BBA | | |
| --- | --- | --- | --- | --- | --- | --- | --- | --- | --- |
| | $\rho_t$ | $MJS$ | $TJS$ | $\rho_t$ | $MJS$ | $TJS$ | $\rho_t$ | $MJS$ | $TJS$ |
| **PESLA-specific** | 0.9341 | 0.0203 | 0.2342 | 0.9014 | 0.0200 | 0.2322 | 0.9179 | 0.0207 | 0.2468 |
| **PESLA-cross** | 0.8583 | 0.0875 | 0.3510 | 0.7014 | 0.0775 | 0.4362 | 0.6665 | 0.1055 | 0.4065 |
| **NeuralMJP-cross** | – | 0.5837 | 0.7724 | – | 0.5303 | 0.6703 | – | 0.4382 | 0.5651 |
| | NTL9 | | | A3D | | | | | |
| | $\rho_t$ | $MJS$ | $TJS$ | $\rho_t$ | $MJS$ | $TJS$ | | | |
| **PESLA-specific** | 0.8867 | 0.0167 | 0.2625 | 0.8186 | 0.0414 | 0.3055 | | | |
| **PESLA-cross** | 0.7443 | 0.1089 | 0.4597 | 0.6235 | 0.2068 | 0.6034 | | | |
| **NeuralMJP-cross** | – | 0.3539 | 0.4525 | – | 0.7340 | 0.7983 | | | |

### D.5 SCALABILITY

To investigate the relationship between the size of the state space and the codebook size, we evaluate the impact of the preset number of codewords $K$ on energy and evolution prediction in protein folding datasets with varying numbers of alpha-C atoms, as shown in Figure 10. As $K$ increases from 10 to 1000, the relative performance of the model improves. For the BBA protein, which has only 28 alpha-C atoms, the prediction accuracy for energy reaches over 90% of the performance observed at $K = 1000$ when $K = 100$. For larger proteins, such as A3D, the model's predictive performance converges at $K = 500$. In fact, protein size increases the complexity of the state space, thereby adding to the modeling challenge. For larger proteins or other systems with complex state spaces, the codebook size needs to be sufficiently large to ensure PESLA's modeling capacity. For the systems studied in this paper, we recommend setting the preset number of codewords $K$ to 1000. For other unfamiliar systems, starting with a relatively large $K$ value is generally advisable.

Furthermore, as analyzed in Appendix F, the time complexity for training and inference grows sublinearly with the increase in $K$. Since only a limited number of codewords are activated in the preset codebook, the size of the energy landscape constructed by PESLA is at most $K$. Consequently, when modeling the state transition distribution over the landscape using the Graph Neural Fokker-Planck equation, the number of codewords to be considered does not necessarily increase linearly with $K$. This design allows users to efficiently explore and select appropriate values for $K$.

### E ABLATION STUDIES

PESLA comprises multiple loss function terms and submodules. Here, we introduce additional ablation studies to elucidate the individual contribution of each component to the overall performance.

As summarized in Section 3.3, the training process involves five loss function terms: $L_{rec}$, $L_{vq}$, $L_{latent}$, $L_{code}$, and $L_{phy}$. The $L_{rec}$ and $L_{vq}$ terms jointly guide the adaptive codebook learning module, while $L_{latent}$ and $L_{code}$ direct the synergistic learning process for energy and evolution prediction. Additionally, the $L_{phy}$ term incorporates physical knowledge to further inform energy estimation. Beyond the essential loss terms, we individually evaluate the performance impact of the auxiliary terms, $L_{latent}$ and $L_{phy}$. The results are presented in Table 6.

We firstly remove the $L_{latent}$ term, training the prediction module solely with the $L_{code}$ term. The results indicate that, without the predictive constraint from the latent space, the accuracy of evolution prediction deteriorates, and the precision of energy estimation is also affected. Next, upon removing the $L_{phy}$ term, PESLA's performance in both energy and evolution prediction declined significantly. This suggests that, although self-supervised learning on the evolution prediction task

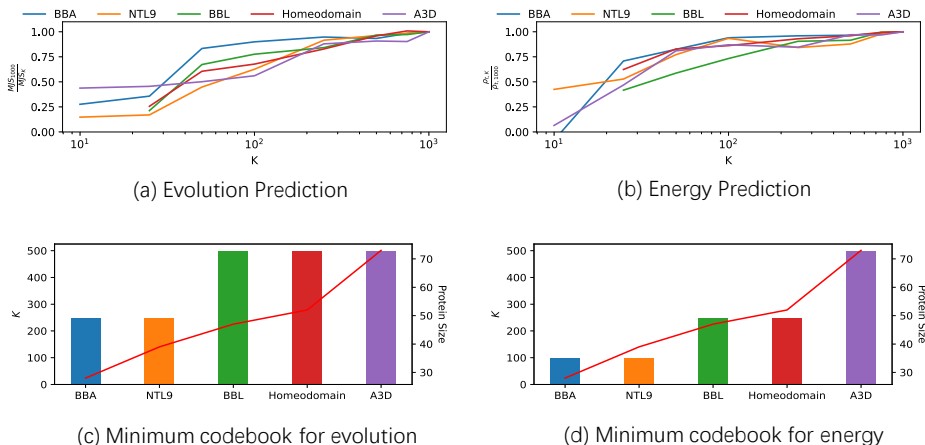

Figure 10: Mean (a) $MJS$ and (b) $\rho_t$ for proteins of different sizes as a function of codebook size $K$, normalized by the metric at $K = 1000$. (c) and (d) report the minimum codebook size required to achieve 90% prediction performance for different proteins. All experiments are run 10 times to obtain mean values.

can drive energy estimation, physical knowledge remains crucial for guiding this joint optimization task effectively.

In Section 3.2, we enhance the model's capability by encoding one-hot probability vectors as initial conditions for the Graph Neural Fokker-Planck equation. Here, we validate this design. By deactivating the encoder $\Phi$ and decoder $\Psi$, we require the neural Fokker-Planck equation to directly model the diffusion of the probability vector. As shown in Table 6, when $\Phi$ and $\Psi$ are deactivated, PESLA's performance in both energy and evolution prediction deteriorates, confirming the importance of the high-dimensional encoded space for effective graph neural diffusion modeling.

Table 6: Ablation study on the loss function and submodule for 2D Prinz Potential and Ecological Evolution. w/o * indicates the absence of the loss function * or module *. All experiments are run 10 times to obtain statistical values.

| | 2D Prinz potential | | | |
|---|---|---|---|---|
| | $\rho_t$ | $\rho_f$ | $MJS$ | $TJS$ |
| PESLA | $0.9290 \pm 0.0342$ | $0.7419 \pm 0.0318$ | $0.1031 \pm 0.0125$ | $0.1796 \pm 0.0234$ |
| w/o $L_{phy}$ | $0.0641 \pm 0.0182$ | $0.003 \pm 0.0928$ | $0.1435 \pm 0.0102$ | $0.2559 \pm 0.0358$ |
| w/o $L_{latent}$ | $0.8089 \pm 0.0672$ | $0.7192 \pm 0.0291$ | $0.1270 \pm 0.0334$ | $0.2010 \pm 0.0327$ |
| w/o $\Phi\&\Psi$ | $0.8994 \pm 0.0477$ | $0.6925 \pm 0.0903$ | $0.1675 \pm 0.0089$ | $0.3535 \pm 0.0122$ |
| | Ecological Evolution | | | |
| | $\rho_t$ | $\rho_f$ | $MJS$ | $TJS$ |
| PESLA | $-0.9067 \pm 0.0100$ | $-0.7582 \pm 0.0241$ | $0.3111 \pm 0.0397$ | $0.3277 \pm 0.0424$ |
| w/o $L_{phy}$ | $-0.0271 \pm 0.0281$ | $-0.002 \pm 0.0817$ | $0.4455 \pm 0.0865$ | $0.4683 \pm 0.0257$ |
| w/o $L_{latent}$ | $-0.8982 \pm 0.0071$ | $-0.6912 \pm 0.0182$ | $0.3228 \pm 0.0441$ | $0.3441 \pm 0.0232$ |
| w/o $\Phi\&\Psi$ | $-0.8980 \pm 0.0075$ | $-0.7018 \pm 0.0202$ | $0.3564 \pm 0.0236$ | $0.4685 \pm 0.0227$ |

## F    COMPUTATIONAL COST

We denote the sample size and the preset number of codewords as $N$ and $K$, respectively. The training process of PESLA consists of two modules: adaptive codebook learning and the graph neural Fokker-Planck equation. The former includes encoding and decoding each sample, as well as codeword matching operations. The computational complexity of encoding and decoding is $\mathcal{O}(N)$, while codeword matching, which involves similarity calculations with each codeword, has a time complexity of $\mathcal{O}(NK)$. In the second module, all computations occur on the codeword topology

with a size of $\mathcal{O}(rK)$, where $r$ is the proportion of activated codewords. Since the encoding, decoding, and diffusion processes for the probability vector, as described in Equations 4 and 5, involve operations over the entire topology, the computational complexity is $\mathcal{O}(rNK)$. Therefore, the overall time complexity during training is $\mathcal{O}(N + NK + rNK) = \mathcal{O}(NK)$.

Once training is complete, the model only needs to retain the activated codewords, resulting in an inference time complexity of $\mathcal{O}(rNK)$. In the worst case, $r = 100\%$, meaning a 100% codeword activation ratio. However, as reported in Section 4.3, for a discrete state space of size 10,000, fewer than 100 codewords are typically sufficient for reliable prediction. Thus, $r$ is usually a low value, making the model's inference cost manageable. Additionally, our encoder $\Xi$ employs an MLP architecture, with a time complexity that scales linearly with the dimension of the observed state.

We evaluate the training time of all algorithms across three datasets, with batch size and epochs uniformly set to 128 and 10, respectively, to ensure fairness. The experiments are conducted on a hardware platform equipped with an Intel i5-14600KF CPU and an NVIDIA RTX 4060Ti GPU. As shown in Figure 11, the total runtime of PESLA's two phases is shorter than that of the optimal baseline, NeuralMJP, reported in the main text. Additionally, PESLA's time bottleneck is clearly concentrated in phase 1 (adaptive codebook learning), with phase 2 (graph neural Fokker-Planck equation) accounting for less than half of the total training time, aligning with the conclusions of previous analysis.

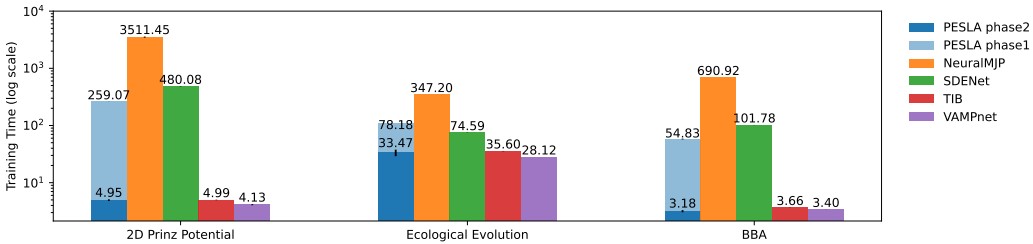

Figure 11: Total training time of all algorithms across the three datasets. All experiments are run 10 times to obtain mean values.

## G  DEGRADED OBSERVATION

In Equation 2 of the main text, the system's intrinsic state evolving on the energy landscape is mapped to the observation space via the observation function $g$, which serves as the input to PESLA. In the main experiments, the observation state $x_t$ retains the primary information of $s_t$; however, this may not hold under certain lossy observation functions. In complex systems modeling, the time-delay embedding method reconstructs the manifold of system's evolution by embedding multi-step trajectories of a high-dimensional system in a limited-dimensional space (Wu et al., 2024). Here, we supplement a set of degraded observation experiments to verify that PESLA can leverage a similar idea, modeling the energy landscape using multi-step historical observations.

Table 7: Energy prediction as a function of lookback steps.

| lookback | 1 | 2 | 3 | 4 | 5 |
|---|---|---|---|---|---|
| $\rho_t$ | 0.7087 | 0.7338 | 0.7916 | 0.8037 | 0.8056 |

Specifically, we applied an observation function $g(x,y) = \begin{bmatrix} \cos(\frac{\pi}{4}) & \sin(\frac{\pi}{4}) \end{bmatrix} \begin{bmatrix} x \\ y \end{bmatrix}$ to the 2D Prinz Potential used in the main text, projecting the 2-dimensional system state coordinates onto the 1-dimensional diagonal of the energy landscape as a degraded observation state. We tested the performance of PESLA's energy prediction as a function of the historical observation step length, as shown in Table 1. The prediction performance is poor when using only a single observation step, as the one-dimensional degraded observation lacks complete information about the energy. As the input

lookback steps increase, the model gains access to the system's past evolution trajectories, improving prediction performance to over 85% of that under lossless observations. The results indicate that degraded observations can be mitigated by incorporating multi-step historical trajectories, aligning with the consensus in the field.

