# OpenReview forum: "Predicting the Energy Landscape of Stochastic Dynamical System via  Physics-informed Self-supervised Learning"
_ICLR.cc/2025/Conference — ICLR 2025 Poster_

### Official Review · Reviewer_iFFC · 2024-10-29

**Soundness:** 3
**Presentation:** 4
**Contribution:** 3
**Rating:** 6
**Confidence:** 3

**Summary:**

This paper proposed a new framework PESLA to estimate the energy landscapes from the evolution trajectories of the system. The framework includes two stage, first it uses an adaptive codebook to obtain a discrete landscape space from the observation trajectories. Second, it uses a graph neural ODE based on the Fokker-Planck equation to predict system evolution.

**Strengths:**

- The paper is clearly written.

- The method is trained in a self-supervised manner, hence it can conduct estimation without knowing true energy landscape.

- The methods were evaluated on three interdisciplinary systems and all demonstrated an improvement compared with SOTA baselines.

**Weaknesses:**

- The author mentioned that “observable evolutionary trajectories typically cover only a limited portion of the vat state space”, however, the paper did not explain how the purposed method perform in the situation when the available data is biased or very sparse.

- The method relies on the assumption that the system is driven by energy and a well-defined low dimensionality energy landscape exists. Therefore, the assumption might limit the applicability, which was not discussed in the paper.

- The method is lack of interpretablity. The training fully relies on the final trajectory prediction errors to avoid knowing true energy landscape, however, it is not discussed how reliable or meaningful of the predicted trajectory and the estimated energy landscape. For example, dose the correlation of the estimated energy landscape and true energy landscape influence the final trajectory prediction?

**Questions:**

- What is the computational complexity of the proposed methods and its comparison to existing methods?

- Are the estimate energy landscape consistent under different hyperparameters?

- Does the granularity of the discretization of the energy landscape influence the accuracy of system evolution estimation?

- Considering that real-world trajectory data is usually noisy and sparse, how robust is the method to such data?

---

> ### Author Response · Authors · 2024-11-21
> **Response to Reviewer iFFC (Part1)**
>
> We would like to sincerely thank Reviewer iFFC for providing a detailed review and insightful suggestions. According to the reviewers' valuable suggestions, we have revised the manuscript and **uploaded the latest version of the PDF**. We endeavored to address all the comments and our reflections are now given below point by point.
>
>
>
> **Q1: Discussion of data bias and sparsity**
>
> A major challenge in data-driven modeling of dynamical systems is that observational trajectories may be biased or overly sparse. In the proposed method, the adaptive codebook learning module coarsens the encoding space through codeword mapping. This approach reduces the impact of limited observational coverage by consolidating system states with similar semantic features into a unified representation. A codeword-controlled region is essentially a continuous local area in the encoding space, within which there are many (infinite) unseen sample points during training. Directly using these unseen points for evolution prediction would result in significant errors. However, when encountering a new observational sample, the encoder projects it into the encoding space, and the adaptive codebook module can match it to the most similar codeword, allowing evolution prediction to be based on the features of this known codeword, thereby improving accuracy.
>
> Furthermore,  we have conducted experimental analysis in $\underline{\text{Appendix D.4 of the revised paper}}$ (Table1 below). Specifically, we performed cross-protein experiments on five protein datasets, using the adaptive codebook learning module trained on other proteins to test on unseen proteins. The results show that, while performance degradation is inevitable due to unobserved regions, the **average transfer performance $\rho_t$ for energy prediction across all proteins reaches over 80%** of the performance of specifically trained version.
>
>
>
> Table1: Comparison of mean $\rho_t$, MJS, and TJS metrics for encoder $\Xi$, decoder $\Omega$, and codebook $C$ trained on specific protein data versus cross-protein data for Homeodomain, BBL, BBA, NTL9, and A3D. All experiments are run 10 times to obtain mean values.
>
> |                    | Homeodomain |        |        | BBL      |        |        | BBA      |        |        |
> | ------------------ | ----------- | ------ | ------ | -------- | ------ | ------ | -------- | ------ | ------ |
> |                    | $\rho_t$    | $MJS$  | $TJS$  | $\rho_t$ | $MJS$  | $TJS$  | $\rho_t$ | $MJS$  | $TJS$  |
> | **PESLA-specific** | 0.9341      | 0.0203 | 0.2342 | 0.9014   | 0.0200 | 0.2322 | 0.9179   | 0.0207 | 0.2468 |
> | **PESLA-cross**    | 0.8583      | 0.0875 | 0.3510 | 0.7014   | 0.0775 | 0.4362 | 0.6665   | 0.1055 | 0.4065 |
> |                    | **NTL9**    |        |        | **A3D**  |        |        |          |        |        |
> |                    | $\rho_t$    | $MJS$  | $TJS$  | $\rho_t$ | $MJS$  | $TJS$  |          |        |        |
> | **PESLA-specific** | 0.8867      | 0.0167 | 0.2625 | 0.8186   | 0.0414 | 0.3055 |          |        |        |
> | **PESLA-cross**    | 0.7443      | 0.1089 | 0.4597 | 0.6235   | 0.2068 | 0.6034 |          |        |        |
>
>
>
> **Q2: Discussion of limitations**
>
> Thanks for your valuable suggestions. This work focuses on estimating the energy landscape of a class of energy-driven evolutionary systems. However, when a system is driven by non-conservative forces,  an energy landscape does not exist, as in the case of motion in viscous fluids. Additionally, inferring energy landscapes becomes more challenging when the landscape is time-varying, such as in cases where climate change alters species fitness. For time-varying landscapes, where $E( * )$ need to adapt to $E( * ,t)$, further model design exploration will be necessary to accommodate these dynamics in the future.
>
>  Overall, we have added  limitations in the $\underline{\text{Conclusion of the revised paper}}$.

---

> ### Author Response · Authors · 2024-11-21
> **Response to Reviewer iFFC (Part2)**
>
> **Q3: Analysis of interpretablity**
>
> Thank you very much for your insightful suggestions.  We have added analysis of the interpretability of the proposed method in the revised paper:
>
> - We experimentally validated the impact of energy estimation on evolution prediction in $\underline{\text{Appendix D.1 of the revised paper}}$ (Table1 below). We replaced the predicted energy with dummy energy to train the evolution prediction module. As the correlation between the dummy energy and true energy decreases, the prediction error for evolution increases. **When the energy correlation drops below 0.5, the evolution prediction error of the graph neural Fokker-Planck equation module gradually exceeds that of the optimal baseline**, highlighting the importance of accurate energy estimation for evolution prediction.
> - We experimentally validated the impact of evolution prediction on energy estimation in $\underline{\text{Appendix E of the revised paper}}$ (Table2 below). Specifically, we conducted ablation experiments by disabling the auxiliary loss term $L_{latent }$ that guides evolution prediction. The results show that **without the guidance for evolution prediction, the correlation of the estimated energy also declines**. This suggests that accurate evolution prediction plays a significant role in energy estimation within our collaborative learning framework.
>
>
>
> Table1: Evolution prediction accuracy as a function of  the correlation coefficient $\rho$ between the dummy energy and the true energy on the 2D Prinz potential. The last two columns report the results of the proposed model (PESLA) and the best baseline (NeuralMJP) as presented in the main text, serving as a reference.
>
> | $\rho$ | 0.9    | 0.7    | 0.5     | 0.3    | 0.1    | PESLA  | NeuralMJP |
> | ------ | ------ | ------ | ------- | ------ | ------ | ------ | --------- |
> | $MJS$  | 0.1058 | 0.1053 | 0.1332  | 0.1283 | 0.1521 | 0.1031 | 0.1463    |
> | $TJS$  | 0.1807 | 0.1865 | 0.23228 | 0.2137 | 0.2589 | 0.1796 | 0.2282    |
>
>
>
> Table2: Ablation study on the loss function and submodule for 2D Prinz Potential and Ecological Evolution. w/o * indicates the absence of the loss function * or module *. All experiments are run 10 times to obtain statistical values.
>
> |                          | $\rho_t$         | $\rho_f$         | MJS             | TJS             |
> | ------------------------ | ---------------- | ---------------- | --------------- | --------------- |
> | **2D Prinz potential**   |                  |                  |                 |                 |
> | PESLA                    | 0.9290 ± 0.0342  | 0.7419 ± 0.0318  | 0.1031 ± 0.0125 | 0.1796 ± 0.0234 |
> | w/o $L_{latent}$         | 0.8089 ± 0.0672  | 0.7192 ± 0.0291  | 0.1270 ± 0.0334 | 0.2010 ± 0.0327 |
> | **Ecological Evolution** |                  |                  |                 |                 |
> | PESLA                    | -0.9067 ± 0.0100 | -0.7582 ± 0.0241 | 0.3111 ± 0.0397 | 0.3277 ± 0.0424 |
> | w/o $L_{latent}$         | -0.8982 ± 0.0071 | -0.6912 ± 0.0182 | 0.3228 ± 0.0441 | 0.3441 ± 0.0232 |
>
>
>
> **Q4: Analysis of computational complexity**
>
> We have added an analysis and experiments on computational complexity in $\underline{\text{Appendix F of the revised paper}}$ to address this question. During the training phase, the main computational bottleneck of the proposed method comes from codeword matching, with overall complexity growing linearly with the sample size and the number of predefined codewords. Once the adaptive codebook module is trained, the model retains only the activated codewords, significantly reducing inference costs. Overall, denoting the sample size as $N$, the predefined number of codewords as $K$, and the token activation rate as $r$, the time complexities for training and testing are $\mathcal{O}(NK)$ and $\mathcal{O}(rNK)$, respectively. Additionally, we compared the training time of all models under the same settings. The experimental results ($\underline{\text{Figure 11 of the revised paper}}$) are consistent with the analytical conclusions, showing that the proposed method’s training time is comparable to that of the baseline.
>
>
> **Q5: Analysis of consistency**
>
> We conducted a consistency analysis of the estimated energy landscapes under different hyperparameter settings and random seeds in $\underline{\text{Appendix D.2 of the revised paper}}$. The results show that for the same protein, models trained with varying codebook sizes or random seeds achieve **an average correlation of over 0.9** in their predictions. This demonstrates that the proposed method is robust to hyperparameter selection, producing consistent results.

---

> ### Author Response · Authors · 2024-11-21
> **Response to Reviewer iFFC (Part3)**
>
> **Q6: Discussion of the granularity of the discretization**
>
> This question involves the sensitivity analysis of the proposed method to hyperparameters. In the original paper, we conducted experiments by setting different numbers of codewords, with results reported in $\underline{\text{Figures 2c and 3c of Sec.4}}$. Additionally, in $\underline{\text{Appendix D.5 of the revised paper}}$, we tested the codeword requirements for proteins of varying sizes. When the codeword count is low, the energy landscape becomes coarsely discretized, making modeling capacity a performance bottleneck. Distinct observational states may become indistinguishable due to an insufficient number of codewords, leading to a decline in prediction quality. As the preset number of codewords increases, the model has sufficient codewords to represent the states of interest, resulting in improved prediction performance. Further increasing the codeword count, however, does not lead to unlimited improvements; rather, the prediction performance gradually converges as the number of codewords grows.
>
>
>
> **Q7: Discussion of the noisy data**
>
> We tested the model's prediction performance under different noise levels in $\underline{\text{Appendix D.3 of the revised paper (Figure 9)}}$. Using the amplitude of the observed state values as a standard, the model **remained stable and did not collapse when noise intensity was below 0.5**, demonstrating sufficient robustness. Such robustness to noise can be attributed to its adaptive codebook learning model, which incorporates a reduced-order approach. By identifying a low-dimensional, compact representation of the original state space, PESLA inherently possesses the ability to filter out uncertainties such as noise-related errors.

---

> ### Author Response · Authors · 2024-11-25
> **Willing to further clarify your remaining concerns**
>
> Dear reviewer,
>
> The discussion period is almost ending. Could you please confirm whether our responses have alleviated your concerns?
>
> For your interest, we (1) discuss the data bias and sparsity; (2) discuss the limitations of the method; (3) add analysis of interpretablity; (4) add analysis of computational complexity; (5) discuss the granularity of the discretization; (6) discuss the noisy data. Please refer to the details in the responses for you.
>
> If you have further comments, we are also very happy to have a discussion. Thank you very much!

---

> ### Comment · Reviewer_iFFC · 2024-11-25
>
> Thank you for addressing my concerns and answering my questions in details. However, regarding the overal limitations of the work, I am unable to confidently raise the score beyond marginally above the acceptance threshold.

---

> > ### Author Response · Authors · 2024-11-26
> >
> > We are delighted that our experiments and theoretical analyses have addressed your questions and concerns. We sincerely appreciate the time you have taken to review our work. Your comments on interpretability and robustness, as well as considerations for noisy or sparse data, have been invaluable in helping us improve the clarity and depth of the discussion. We hope these efforts effectively underscore the importance and potential of our contribution.
> >
> > We understand that you have some lingering reservations about the overall limitations of the work. However, we believe this study represents a significant step forward in predicting energy landscapes without relying on true energy supervision. Our method is well-suited for estimating existing energy landscapes and holds great potential for discovering previously unknown ones.
> >
> > If there are any specific concerns regarding the limitations that you feel remain unaddressed, we would be eager to hear your thoughts. Your feedback continues to be invaluable in shaping and strengthening this work.

---

### Official Review · Reviewer_jDjA · 2024-10-30

**Soundness:** 3
**Presentation:** 3
**Contribution:** 3
**Rating:** 6
**Confidence:** 3

**Summary:**

Estimating the energy landscape is challenging because obtaining direct energy measurements is costly. To bypass this process, the paper proposes a physics-informed self-supervised learning approach, where the energy landscape is estimated from historical evolution trajectories instead of direct energy signals. This method uses discrete codebook embeddings, assuming that energy landscapes generally have low intrinsic dimensionality. Additionally, a physics-informed graph neural Fokker-Planck architecture and physics-inspired regularization are employed to predict system states more accurately.

**Strengths:**

The paper begins with a strong motivation, highlighting the challenges in obtaining direct energy values. Additionally, it provides a solid rationale for incorporating discrete codebook embeddings into their method, based on the inherently low dimensionality of energy landscapes.

**Weaknesses:**

1. The ablation studies in this paper seem too limited. Since multiple modules are introduced in their method, additional experiments are needed to clarify the individual contributions of each component to the overall performance. For instance, it would be helpful to examine the effect of using discrete versus continuous embeddings. Similarly, given that five types of losses are included, an analysis of each loss’s specific impact would provide valuable insights. If these results were included, I would increase my rating.

2. Regarding  $L_{phy}$, could you explain in more detail why this loss is referred to as “physics-inspired”? How it differs from standard machine learning loss terms?

3. There are five types of losses ( L_{reconstruct}, L_{vq}, L_{latent}, L_{code}, L_{phy}) in total. Could you clarify the exact meaning of the terms $\mathbf{p}$ and $\mathbf{q}$ in each loss?

**Questions:**

See Weaknesses

---

> ### Author Response · Authors · 2024-11-21
> **Response to Reviewer jDjA (Part1)**
>
> Many thanks to Reviewer jDjA for providing thorough detailed comments. According to the reviewers' valuable suggestions, we have revised the manuscript and **uploaded the latest version of the PDF**. We endeavored to address all the comments and our reflections are now given below point by point.
>
>
>
> **Q1: Analysis of ablation study**
>
> Thank you very much for your insightful suggestions on this work. We have added experiments in $\underline{\text{Appendix E of the revised paper}}$ (Table2 below), where we analyze the role of each loss function term and validate the impact of the auxiliary loss function terms and submodules. Specifically, we:
>
> - Disabled the loss term $L_{latent}$ guiding trajectory prediction and observed a decrease in prediction accuracy, along with a decline in energy estimation precision.
> - Disabled the loss term $L_{phy}$ guiding energy estimation, which made the joint optimization of energy and trajectory prediction more challenging.
> - Disabled the encoder $\Phi$ and decoder $\Psi$​ in the Graph Neural Fokker-Planck equation for the probability vector, resulting in a significant increase in trajectory prediction error and impaired energy estimation.
>
> These results validate the **contributions and necessity** of the proposed loss function terms and modules.
>
> Table2: Ablation study on the loss function and submodule for 2D Prinz Potential and Ecological Evolution. w/o * indicates the absence of the loss function * or module *. All experiments are run 10 times to obtain statistical values.
>
> |                          | $\rho_t$         | $\rho_f$         | MJS             | TJS             |
> | ------------------------ | ---------------- | ---------------- | --------------- | --------------- |
> | **2D Prinz potential**   |                  |                  |                 |                 |
> | PESLA                    | 0.9290 ± 0.0342  | 0.7419 ± 0.0318  | 0.1031 ± 0.0125 | 0.1796 ± 0.0234 |
> | w/o $L_{phy}$            | 0.0641 ± 0.0182  | 0.003 ± 0.0928   | 0.1435 ± 0.0102 | 0.2559 ± 0.0358 |
> | w/o $L_{latent}$         | 0.8089 ± 0.0672  | 0.7192 ± 0.0291  | 0.1270 ± 0.0334 | 0.2010 ± 0.0327 |
> | w/o $\Phi$ & $\Psi$      | 0.8994 ± 0.0477  | 0.6925 ± 0.0903  | 0.1675 ± 0.0089 | 0.3535 ± 0.0122 |
> | **Ecological Evolution** |                  |                  |                 |                 |
> | PESLA                    | -0.9067 ± 0.0100 | -0.7582 ± 0.0241 | 0.3111 ± 0.0397 | 0.3277 ± 0.0424 |
> | w/o $L_{phy}$            | -0.0271 ± 0.0281 | -0.002 ± 0.0817  | 0.4455 ± 0.0865 | 0.4683 ± 0.0257 |
> | w/o $L_{latent}$         | -0.8982 ± 0.0071 | -0.6912 ± 0.0182 | 0.3228 ± 0.0441 | 0.3441 ± 0.0232 |
> | w/o $\Phi$ & $\Psi$      | -0.8980 ± 0.0075 | -0.7018 ± 0.0202 | 0.3564 ± 0.0236 | 0.4685 ± 0.0227 |
>
>
>
> Additionally, we replaced discrete embeddings with continuous embeddings. Although modeling transition probabilities in a continuous state space is not directly compatible with the proposed Graph Neural Fokker-Planck Equation, we used RNN to model the transition probabilities in the encoded space for comparison between continuous and discrete embeddings. The test results on the Ecological Evolution dataset are as follows:
>
> |                       | MJS    | TJS    |
> | --------------------- | ------ | ------ |
> | continuous embeddings | 0.5832 | 0.6920 |
> | PESLA                 | 0.3111 | 0.3277 |
>
> The predictive performance of continuous embeddings is significantly worse, primarily because they lose many of the advantages designed into our method. The discrete codebook learning aligns with the low-dimensional landscape where long-term evolution of the dynamical system unfolds (see Sec 3.1 of the paper). Continuous embedding methods face generalization issues when encountering samples not present in the training set [4,5]. In contrast, our discrete embedding approach mitigates the impact of limited observational coverage by coarsening system states with similar semantic features into a unified representation. Additionally, the dimensionality reduction achieved through discrete mapping filters out local uncertainties introduced by random noise in the raw observational data (see $\underline{\text{Appendix D.3 of the revised paper}}$), ensuring that the predictor focuses on the essential shape of the dynamics. This is crucial for modeling stochastic dynamical systems.

---

> ### Author Response · Authors · 2024-11-21
> **Response to Reviewer jDjA (Part2)**
>
> **Q2: Explanation of “physics-inspired”**
>
> The loss term $L_{phy}$ is formally represented as a KL divergence but essentially serves to introduce a physical prior. Specifically, in the absence of direct supervisory signals to guide energy estimation, the joint optimization of energy estimation and trajectory prediction becomes challenging. Therefore, we aim to incorporate physical prior knowledge to partially constrain the optimization direction. This constraint should be sufficiently general to apply across various disciplinary scenarios, rather than being limited to a specific system.
>
> In statistical physics, a system's energy is used to quantify the probability distribution of states, known as the Boltzmann distribution. This implies a direct relationship between the frequency of a state being observed and its energy level, which can help guide the optimization direction for energy estimation. Furthermore, although different fields may define this concept differently—such as 'fitness' in genetics or 'potential energy' in proteins—prior studies have shown consistency between these concepts and the notion of energy in statistical physics [1,2,3]. Thus, we incorporate the Boltzmann distribution from statistical physics as a guiding principle in the unsupervised task of energy estimation, promoting the joint optimization of energy and trajectory prediction.
>
>
> **Q3: Clarification of Terminology**
>
> Thanks for your scientific rigor. Here, we clarify the notation used in each loss term:
>
> - $L _ {reconstruct}=-\log\mathbf{q} _ {\Xi,\Omega,C}(x)$, where $\mathbf{q} _ {\Xi,\Omega,C}$ represents the probability distribution generated by the encoder $\Xi$, decoder $\Omega$, and codebook $C$. In the original text, $\mathbf{q} _ {\Xi,\Omega,C}$ was mistakenly written as $\mathbf{q} _ {\Xi,\Psi,C}$; this has been corrected.
> - $L _ {latent}=||\Phi(\mathbf{p}(c _ {t+\Delta t})) - \Psi(H(t+\Delta t))||$, where $\mathbf{p} _ {c _ {t+\Delta t}}$ represents the target distribution of states at the time  $t+\Delta t$ in the evolution prediction task.
> - $L _ {code}=-\mathbf{p}(c _ {t+\Delta t})\log{\mathbf{q}(c _ {t+\Delta t})}$, where $\mathbf{p} _ {c _ {t+\Delta t}}$ represents the target distribution of states at the time $t+\Delta t$. $\mathbf{q} _ {c _ {t+\Delta t}}$ represents the model-predicted distribution of states at time $t+\Delta t$.
> - $L _ {phy} = D _ {\text{KL}}(\mathbf{p} \| \mathbf{q}) = \sum _ {i=0}^{K} \mathbf{p}(c _ i) \log \left(\frac{\mathbf{p}(c _ i)}{\mathbf{q}(c _ i)}\right)$, where $\mathbf{p}(c _ i)$ represents the empirical distribution of the codeword $c _ i$ in the training of evolution prediction. $\mathbf{q}(c _ i)$ represents the Boltzmann distribution derived from the estimated energy of the codeword $c _ i$.
>
>
> Ref:
>
> [1] Sella, Guy, and Aaron E. Hirsh. "The application of statistical physics to evolutionary biology." *Proceedings of the National Academy of Sciences* 102.27 (2005): 9541-9546.
>
> [2] Noé, Frank, et al. "Boltzmann generators: Sampling equilibrium states of many-body systems with deep learning." *Science* 365.6457 (2019): eaaw1147.
>
> [3] Thirumalaiswamy, Amruthesh, Robert A. Riggleman, and John C. Crocker. "Exploring canyons in glassy energy landscapes using metadynamics." *Proceedings of the National Academy of Sciences* 119.43 (2022): e2210535119.
>
> [4] Mao, Chengzhi, et al. "Discrete Representations Strengthen Vision Transformer Robustness." *International Conference on Learning Representations*.
>
> [5] Yu, Lijun, et al. "Language Model Beats Diffusion-Tokenizer is key to visual generation." *The Twelfth International Conference on Learning Representations*.

---

> ### Comment · Reviewer_jDjA · 2024-11-22
>
> Thank you for addressing my questions thoroughly. Because most of my concerns have been resolved, I will raise my score. However, as I am not a complete expert in this field, I am unable to confidently champion the paper beyond a weak accept.

---

> > ### Author Response · Authors · 2024-11-26
> >
> > We are glad that our experiments and theoretical analyses have addressed your questions and concerns. We sincerely appreciate the time and effort you have dedicated to reviewing our work, which has greatly helped us refine and improve it.

---

### Official Review · Reviewer_ciD9 · 2024-11-02

**Soundness:** 3
**Presentation:** 3
**Contribution:** 3
**Rating:** 5
**Confidence:** 3

**Summary:**

The authors propose an algorithm for discovering the energy function that may explain data trajectories assumed to be driven by the gradient of the energy plus noise. Their model learns a partitioning of the space with one discrete symbol per region and an associated energy value, along with a decoder (and associated encoder) mapping to the observed continuous space. They assume that the dynamics is driven by the energy difference between nearby partitions and Markovian transitions between regions. The model is evaluated on three tasks and compared with several baselines.

**Strengths:**

This is an interesting learning problem, also requiring to learn a latent explanatory space for sampled continuous time trajectories. The authors compare against multiple baselines across everal environments. The results appear encouraging.

**Weaknesses:**

(1) I do not see how such an approach can generalize outside of the visited regions, i.e., the cluster centers corresponding to the codebook entries; in general, the most interesting energy functions can have a very large (if not exponential) number of energy wells, and one needs to generalize from the visited wells to new ones. The reason is that many factors may be composed that give rise to stable solutions. Hence a multivariate representation of the discrete identities must be constructed (think about language, which is discrete but allows generating a huge number of legal combinations of words).

(2) It is not clear to me that g in general will preserve all information about the energy of the state, so it may be necessary for the encoder (from observations to states) to look at the whole past trajectory in order to make a probabilistic guess about the state proxy.

(3) the main model (sec 3.2) is not sufficiently understandable and could use more motivation for its different parts. Why these equations in particular (4 and 5).

(4) The experimental setup seems to lack comparisons with *published results* from earlier work, which makes it difficult to know if the comparisons are fair.

(5) The test set includes only trajectories from the same system for which a training trajectory is given, i.e., there is no form of generalization to new systems (e.g., new molecules).

**Questions:**

* to address (3) above, please provide better justifications for the main model and the associated training losses

* to address (4) and (5), please compare with published results on recovering the energy and fitting new trajectories of new molecules or systems

---

> ### Author Response · Authors · 2024-11-21
> **Response to Reviewer ciD9 (Part1)**
>
> Many thanks to Reviewer ciD9 for providing thorough insightful comments. According to the reviewers' valuable suggestions, we have revised the manuscript and **uploaded the latest version of the PDF**. We endeavored to address all the comments and our reflections are now given below point by point.
>
>
>
> **Q1: Discussion of generalization**
>
> This issue typically arises when observational data is biased or extremely sparse, which is an inherent challenge for data-driven modeling of dynamical systems. In practice, the available observation trajectories may not fully cover all the 'energy wells,' particularly localized wells with limited coverage. However, the proposed adaptive codebook learning module can match unseen states to the codeword whose features are most similar, grouping states with similar energy characteristics. For example, in a large energy well containing multiple sub-wells, even if certain sub-wells are unseen during training, the proposed model can at least capture the information of the overarching energy well. Furthermore, as you mentioned, each codeword in the proposed method does have a multivariate representation, sufficiently characterizing the local state features within its region. To validate this generalization capability, we conducted cross-protein dataset experiments in $\underline{\text{Appendix D.4 of the revised paper}}$ (Table1 below). We trained the adaptive codebook learning module on other proteins and tested it on unseen proteins. The **average transfer performance $\rho_t$​ for energy prediction across all proteins reaches over 80%** of the performance of specifically trained version, demonstrating the codebook learning module’s capability to generalize to out-of-distribution regions.
>
>
>
> Table1: Comparison of mean $\rho_t$, MJS, and TJS metrics for encoder $\Xi$, decoder $\Omega$, and codebook $C$ trained on specific protein data versus cross-protein data for Homeodomain, BBL, BBA, NTL9, and A3D. All experiments are run 10 times to obtain mean values.
>
> |                    | Homeodomain |        |        | BBL      |        |        | BBA      |        |        |
> | ------------------ | ----------- | ------ | ------ | -------- | ------ | ------ | -------- | ------ | ------ |
> |                    | $\rho_t$    | $MJS$  | $TJS$  | $\rho_t$ | $MJS$  | $TJS$  | $\rho_t$ | $MJS$  | $TJS$  |
> | **PESLA-specific** | 0.9341      | 0.0203 | 0.2342 | 0.9014   | 0.0200 | 0.2322 | 0.9179   | 0.0207 | 0.2468 |
> | **PESLA-cross**    | 0.8583      | 0.0875 | 0.3510 | 0.7014   | 0.0775 | 0.4362 | 0.6665   | 0.1055 | 0.4065 |
> |                    | **NTL9**    |        |        | **A3D**  |        |        |          |        |        |
> |                    | $\rho_t$    | $MJS$  | $TJS$  | $\rho_t$ | $MJS$  | $TJS$  |          |        |        |
> | **PESLA-specific** | 0.8867      | 0.0167 | 0.2625 | 0.8186   | 0.0414 | 0.3055 |          |        |        |
> | **PESLA-cross**    | 0.7443      | 0.1089 | 0.4597 | 0.6235   | 0.2068 | 0.6034 |          |        |        |
>
>
>
> **Q2: Discussion of observation function g**
>
> Thanks for this insightful question. In the study of complex system modeling, delay embedding theory demonstrates that manifolds of system evolution can be reconstructed from multi-step observation trajectories [1]. In the three experimental scenarios presented in the main text of our paper, the observation function retains all information about the state energy, allowing us to use single-step observations as input. We have added supplementary experiments on degraded observations in $\underline{\text{Appendix G of the revised paper}}$ (Table2 below). Specifically, we applied an observation function $g(x, y) = \begin{bmatrix} \cos(\frac{\pi}{4}) & \sin(\frac{\pi}{4}) \end{bmatrix} \begin{bmatrix} x \\ y \end{bmatrix}^T$ to the 2D Prinz Potential used in the main text, projecting the 2-dimensional system state coordinates onto the 1-dimensional diagonal of the energy landscape as a degraded observation state. The results show that when the observation state lacks complete information about the energy, our model can improve energy prediction by increasing the number of input observation steps. As the input lookback steps increase, the model gains access to the system's past evolution trajectories, improving prediction performance to **over 85%** of that under lossless observations.
>
>
>
> Table2: Energy prediction as a function of lookback steps.
>
> | lookback | 1      | 2      | 3      | 4      | 5      |
> | -------- | ------ | ------ | ------ | ------ | ------ |
> | $\rho_t$ | 0.7087 | 0.7338 | 0.7916 | 0.8037 | 0.8056 |

---

> ### Author Response · Authors · 2024-11-21
> **Response to Reviewer ciD9 (Part2)**
>
> **Q3: Explanation of Sec. 3.2**
>
> We greatly appreciate your valuable suggestions. In $\underline{\text{Sec. 3.2 of the revised paper}}$, we have expanded on the motivation behind employing the graph neural Fokker-Planck equation to model the energy-driven evolution of system states within the landscape space.
>
> Specifically, in a one-dimensional probability distribution, the information is limited to the immediate state of the node. In contrast, by using a graph convolutional network (GCN) to encode the probability vector into a high-dimensional space, we can effectively capture richer relational information between nodes, leveraging the GCN’s ability to aggregate and integrate neighborhood features. This relational structure is crucial in the evolution modeled by  graph neural Fokker-Planck equation, as the change in a node’s state often depends on its neighbors. A higher-dimensional representation allows the model to more accurately reflect these dependencies. Moreover, the high-dimensional space allows the model to leverage the nonlinear fitting capabilities of neural networks, enabling it to capture complex patterns in the energy landscape, thereby improving prediction accuracy.
>
> Additionally, in $\underline{\text{Appendix E of the revised paper}}$, we conducted detailed **ablation** experiments to analyze the contributions of the probability encoding module and each loss function.
>
>
>
> **Q4: Comparison of experimental setup**
>
> In dynamical system modeling research, it is common and well-accepted in the community to use different trajectories generated by the same system for training and testing [2,3,4]. We comprehensively evaluate the proposed model's performance in evolution and energy prediction using both JS divergence and Pearson correlation coefficient, consistent with prior works [2,5,6]. Additionally, we apply the same evaluation settings and average results over multiple runs for all models to ensure fairness in the evaluation.
>
>
>
> Ref:
>
> [1] Wu, Tao, et al. "Predicting multiple observations in complex systems through low-dimensional embeddings." *Nature Communications* 15.1 (2024): 2242.
>
> [2] Federici, Marco, et al. "Latent Representation and Simulation of Markov Processes via Time-Lagged Information Bottleneck." *The Twelfth International Conference on Learning Representations*.
>
> [3] Seifner, Patrick, and Ramsés J. Sánchez. "Neural Markov jump processes." *International Conference on Machine Learning*. PMLR, 2023.
>
> [4] Mardt, Andreas, et al. "VAMPnets for deep learning of molecular kinetics." *Nature communications* 9.1 (2018): 5.
>
> [5] Faure, Andre J., et al. "Mapping the energetic and allosteric landscapes of protein binding domains." *Nature* 604.7904 (2022): 175-183.
>
> [6] Yang, Huan, Zhaoping Xiong, and Francesco Zonta. "Construction of a deep neural network energy function for protein physics." *Journal of Chemical Theory and Computation* 18.9 (2022): 5649-5658.

---

> ### Author Response · Authors · 2024-11-25
> **Willing to further clarify your remaining concerns**
>
> Dear reviewer,
>
> The discussion period is almost ending. Could you please confirm whether our responses have alleviated your concerns?
>
> For your interest, we (1) discuss the generalization of the method; (2) discuss the observation function g; (3) add explanation of Sec. 3.2; (4) add comparison of experimental setup. Please refer to the details in the responses for you.
>
> If you have further comments, we are also very happy to have a discussion. Thank you very much!

---

> ### Author Response · Authors · 2024-11-30
> **Request for Feedback on Rebuttal**
>
> As the rebuttal period is drawing to a close, we are concerned about the lack of communication with you. Could you please see whether our responses will influence your ratings? All the other reviewers gave our paper a rating above acceptance threshold. You are the sole reviewer who has expressed reservations, and we take your concerns very seriously. We have addressed all the issues you raised in our rebuttal, but have not yet received any feedback from you. We respectfully request your prompt response.

---

### Official Review · Reviewer_c7fR · 2024-11-03

**Soundness:** 3
**Presentation:** 2
**Contribution:** 3
**Rating:** 6
**Confidence:** 4

**Summary:**

This paper proposes to infer the energy landscape of complex, stochastic dynamical systems. A distinguishing feature is the adoption of vector quantization techniques for dimension reduction. Dynamical information was inferred via a graph neural Fokker-Planck equation method. A regularization term was introduced to constrain the long-term prediction behavior between the empirical distribution p and the Boltzmann distribution. The novelty of bringing together these ideas is appreciated. Experiments were performed on 3 tests: 2d Prinz potential, ecological evolution, and fast-folding peptides. The method part of the paper is not very clear and somewhat difficult to follow. There is no ablation tests, so it's hard to see what the various loss terms or model designs offer. There is no discussion of the limitations of the method.

**Strengths:**

The idea of bringing together several novel methods for modeling stochastic dynamical systems is very interesting. Three different examples cover different application areas and add to the credibility of the method. The capability to learn unknown energy landscape from stochastic dynamical systems is a potentially high-impact contribution.

**Weaknesses:**

* The description of the overall model is not clear to me. At least not easy to follow. For example, the encoder, decoder, Phi, Psi, Xi,... were defined in different places and some were not clearly defined. What space and dimension does Phi map from and to? Please make it easier to follow. And the dimension of latents such as H is not clear. Please collect the equations in a coherent way, e.g. all the different losses, and what are the losses sampled over?

* Overall, the writing of the paper is too hand-waving. As a referee I don't like guesswork.

**Questions:**

* Please discuss the issue of transferrability. Can the trained model be transferred to unseen structures? Was a separate model trained for each protein? Which of the sub-modules, encoder, decoder, etc, can be reused?

* Scalability. To model a bigger protein, how large does the codebook need to be? What about training and inference cost for longer sequences?

* Spell out KNN (k-nearest neighbor?)

* How are encoder Ξ, decoder Ω implemented? GCNN was mentioned but I still have no idea what's in the model.

* There is no ablation tests, so it's hard to see what the various loss terms or model designs offer.

* There is no discussion of the limitations of the method.

* Consider adding some explanation of the graph FPE method. FPE models the evolution of the probability p, not its encoding Phi(p) as in eq (4). I suppose it is explained in the cited reference. But since it is a central piece of this paper, please explain how this works.

---

> ### Author Response · Authors · 2024-11-21
> **Response to Reviewer c7fR (Part1)**
>
> Many thanks to Reviewer c7fR for providing a detailed review and insightful questions. According to the reviewers' valuable suggestions, we have revised the manuscript and **uploaded the latest version of the PDF**. We endeavored to address all the comments and our reflections are now given below point by point.
>
>
>
> **Q1: Description and implementation of the overall model**
>
> Thanks for your valuable suggestions. We have added new paragraphs and an appendix section to summarize each module:
>
> - Summarized the trainable parameters in the first paragraph of $\underline{\text{Section 3.3 of the revised paper}}$.
> - Listed all modules and parameter shapes in $\underline{\text{Appendix A.2 of the revised paper}}$.
> - Summarized the function of each loss term in $\underline{\text{Appendix E of the revised paper}}$.
>
> While other reviewers, such as reviewer iFFC, have praised the writing of this paper, we sincerely appreciate your suggestions, which have helped us further enhance the clarity of each module’s presentation, making this work more accessible to a broader audience.
>
>
>
> **Q2: Analysis of transferability**
>
> In the original version of the paper, we trained separate models for each protein. We have added transfer experiments on five proteins in $\underline{\text{Appendix D.4 of the revised paper}}$ (Table1 below). We demonstrate that the **encoder $\Xi$, decoder $\Omega$, and codebook $C$ can be reused** across different proteins. By training on cross-protein data, these modules are expected to learn to capture general representations and project diverse protein states into a unified latent space. Notably, using the encoder, decoder, and codebook trained on other proteins, the **average transfer performance $\rho_t$ for energy prediction across all proteins reaches over 80%** of the performance of specifically trained version.
>
>
>
> Table1: Comparison of mean $\rho_t$, MJS, and TJS metrics for encoder $\Xi$, decoder $\Omega$, and codebook $C$ trained on specific protein data versus cross-protein data for Homeodomain, BBL, BBA, NTL9, and A3D. All experiments are run 10 times to obtain mean values.
>
> |                    | Homeodomain |        |        | BBL      |        |        | BBA        |        |        |
> | ------------------ | ----------- | ------ | ------ | -------- | ------ | ------ | ---------- | ------ | ------ |
> |                    | $\rho_t$    | $MJS$  | $TJS$  | $\rho_t$ | $MJS$  | $TJS$  | \(\rho_t\) | $MJS$  | $TJS$  |
> | **PESLA-specific** | 0.9341      | 0.0203 | 0.2342 | 0.9014   | 0.0200 | 0.2322 | 0.9179     | 0.0207 | 0.2468 |
> | **PESLA-cross**    | 0.8583      | 0.0875 | 0.3510 | 0.7014   | 0.0775 | 0.4362 | 0.6665     | 0.1055 | 0.4065 |
> |                    | **NTL9**    |        |        | **A3D**  |        |        |            |        |        |
> |                    | $\rho_t$    | $MJS$  | $TJS$  | $\rho_t$ | $MJS$  | $TJS$  |            |        |        |
> | **PESLA-specific** | 0.8867      | 0.0167 | 0.2625 | 0.8186   | 0.0414 | 0.3055 |            |        |        |
> | **PESLA-cross**    | 0.7443      | 0.1089 | 0.4597 | 0.6235   | 0.2068 | 0.6034 |            |        |        |
>
>
>
> **Q3: Analysis of scalability**
>
> We have added experiments in $\underline{\text{Appendix D.5 of the revised paper}}$ to address this question. In these experiments, the prediction performance for all proteins improves as the codebook size $K$ increases, approaching convergence before reaching $K=1000$. Using $K=1000$ as a benchmark, we found that the minimum codebook size required to achieve 90% performance increases with protein size ($\underline{\text{Figure 10}}$). However, for the proteins used in this study, **a codebook size of $K=500$ is sufficient to meet all modeling requirements**. Additionally, for unfamiliar systems, we recommend starting with a larger $K$ value, as the proposed method can automatically utilize only the necessary subset of codewords, remaining computationally efficient.
>
> We also included a computational cost analysis in $\underline{\text{Appendix F of the revised paper}}$. This analysis shows that the time complexity of the proposed method scales **linearly** with the sequence length.

---

> > ### Comment · Reviewer_c7fR · 2024-11-25
> >
> > I appreciate the effort of the authors in addressing my questions. The most decisive factor in this round is the added transferability tests. I have increased my rating. Note that there are many other more important metrics like extensive tests that are needed to really justify the claimed success for a generally applicable method, but those can wait till the follow-up to be fair. So my new rating is just above the acceptance threshold.

---

> > > ### Author Response · Authors · 2024-11-26
> > >
> > > We are delighted that the additional experiments and theoretical analyses have addressed your concerns. Thank you once again for your valuable suggestions, which have significantly helped us improve the clarity and robustness of this work.
> > >
> > > As you mentioned, this work introduces a method for inferring the energy landscapes of stochastic dynamical systems without relying on true energy supervision signals. It lays the foundation for zero-shot predictions across diverse landscapes, which is a significant and promising topic for future research.
> > >
> > > Your constructive feedback has been invaluable in elevating the quality of our submission, and we sincerely appreciate your thoughtful engagement.

---

> ### Author Response · Authors · 2024-11-21
> **Response to Reviewer c7fR (Part2)**
>
> **Q4: Analysis of ablation study**
>
> We have added experiments in $\underline{\text{Appendix E of the revised paper}}$ (Table2 below) to address this question. For each  auxiliary loss function term or submodule, we:
>
> - Disabled the loss term $L_{latent}$ guiding trajectory prediction and observed a decrease in prediction accuracy, along with a decline in energy estimation precision.
> - Disabled the loss term $L_{phy}$ guiding energy estimation, which made the joint optimization of energy and trajectory prediction more challenging.
> - Disabled the encoder $\Phi$ and decoder $\Psi$ in the Graph Neural Fokker-Planck equation for the probability vector, resulting in a significant increase in trajectory prediction error and impaired energy estimation.
>
> These results validate the **contributions and necessity** of the proposed loss function terms and modules.
>
>
>
> Table2: Ablation study on the loss function and submodule for 2D Prinz Potential and Ecological Evolution. w/o * indicates the absence of the loss function * or module *. All experiments are run 10 times to obtain statistical values.
>
> |                          | $\rho_t$         | $\rho_f$         | MJS             | TJS             |
> | ------------------------ | ---------------- | ---------------- | --------------- | --------------- |
> | **2D Prinz potential**   |                  |                  |                 |                 |
> | PESLA                    | 0.9290 ± 0.0342  | 0.7419 ± 0.0318  | 0.1031 ± 0.0125 | 0.1796 ± 0.0234 |
> | w/o $L_{phy}$            | 0.0641 ± 0.0182  | 0.003 ± 0.0928   | 0.1435 ± 0.0102 | 0.2559 ± 0.0358 |
> | w/o $L_{latent}$         | 0.8089 ± 0.0672  | 0.7192 ± 0.0291  | 0.1270 ± 0.0334 | 0.2010 ± 0.0327 |
> | w/o $\Phi$ & $\Psi$      | 0.8994 ± 0.0477  | 0.6925 ± 0.0903  | 0.1675 ± 0.0089 | 0.3535 ± 0.0122 |
> | **Ecological Evolution** |                  |                  |                 |                 |
> | PESLA                    | -0.9067 ± 0.0100 | -0.7582 ± 0.0241 | 0.3111 ± 0.0397 | 0.3277 ± 0.0424 |
> | w/o $L_{phy}$            | -0.0271 ± 0.0281 | -0.002 ± 0.0817  | 0.4455 ± 0.0865 | 0.4683 ± 0.0257 |
> | w/o $L_{latent}$         | -0.8982 ± 0.0071 | -0.6912 ± 0.0182 | 0.3228 ± 0.0441 | 0.3441 ± 0.0232 |
> | w/o $\Phi$ & $\Psi$      | -0.8980 ± 0.0075 | -0.7018 ± 0.0202 | 0.3564 ± 0.0236 | 0.4685 ± 0.0227 |
>
>
>
> **Q5: Discussion of limitations of the method**
>
> Thanks for your valuable suggestions. This work focuses on estimating the energy landscape of a class of energy-driven evolutionary systems. However, when a system is driven by non-conservative forces,  an energy landscape does not exist, as in the case of motion in viscous fluids. Additionally, inferring energy landscapes becomes more challenging when the landscape is time-varying, such as in cases where climate change alters species fitness. For time-varying landscapes, where $E( * )$ need to adapt to $E( * ,t)$, further model design exploration will be necessary to accommodate these dynamics in the future.
>
>  Overall, we have added  limitations in the $\underline{\text{Conclusion of the revised paper}}$.
>
>
>
> **Q6: Explanation of Graph Neural Fokker-Planck Equation**
>
> Thank you very much for your insightful suggestions. In $\underline{\text{Sec. 3.2 of the revised paper}}$, we have provided a more detailed explanation of the motivation behind using the graph neural Fokker-Planck equation to model the energy-driven evolution of system states in the landscape space, along with further insights on the encoding of probability vectors.
>
> Specifically, in a one-dimensional probability distribution, the information is limited to the immediate state of the node. In contrast, by using a graph convolutional network (GCN) to encode the probability vector into a high-dimensional space, we can effectively capture richer relational information between nodes, leveraging the GCN’s ability to aggregate and integrate neighborhood features. This relational structure is crucial in the evolution modeled by  graph neural Fokker-Planck equation, as the change in a node’s state often depends on its neighbors. A higher-dimensional representation allows the model to more accurately reflect these dependencies. Moreover, the high-dimensional space allows the model to leverage the nonlinear fitting capabilities of neural networks, enabling it to capture complex patterns in the energy landscape, thereby improving prediction accuracy.
>
> Additionally, in $\underline{\text{Appendix E of the revised paper}}$, we have included ablation experiments on encoding probability vectors to validate the **necessity** of this approach.

---

> ### Author Response · Authors · 2024-11-25
> **Willing to further clarify your remaining concerns**
>
> Dear reviewer,
>
> The discussion period is almost ending. Could you please confirm whether our responses have alleviated your concerns?
>
> For your interest, we (1) clarify the description and implementation of the overall model and loss terms; (2) add analysis on transferability; (3) add analysis on scalability; (4) add analysis on ablation study; (5) discuss the limitations of the method; (6) add  explanation of graph neural Fokker-Planck equation. Please refer to the details in the responses for you.
>
> If you have further comments, we are also very happy to have a discussion. Thank you very much!

---

### Author Response · Authors · 2024-12-02
**General Comments by Authors**

# General Comments by Authors

We sincerely thank all the reviewers for their insightful reviews and valuable comments, which are instructive for us to improve our paper further.

The reviewers generally believe that this paper addresses an important problem within the community, acknowledging that the proposed method is "**novel**", "**a potentially high-impact contribution**", and "**well-supported by solid rationale**",  the paper "**has  strong motivation**", "**addresses an interesting learning problem**", and "**is  clearly written**", and the experiments "**cover a wide range of application domains**", "**are cross-disciplinary**", and "**enhance the credibility of the method**".

The reviewers also raised insightful and constructive concerns. We made every effort to address each of the reviewers’ points by providing sufficient evidence and the required results. Below are the updates and responses to all of the reviewers' suggestions and concerns.

1. **Robustness of experiments**

   We provided results demonstrating the consistency of the proposed method under different hyperparameters. Additionally, we included experiments with varying noise strengths to support the claim of the method's robustness. These experiments address Reviewer **iFFC**'s concerns.

2. **Analysis of transferability**

   We added cross-system transfer experiments to assess the model's generalization performance when available data is sparse or biased, addressing the questions raised by Reviewer **C7fR** and **iFFC**.

3. **Analysis of scalability**

   We provided an analysis of the model's computational cost as it varies with data size and codeword count, and added experiments to examine how the required codeword count increases with system scale. These experiments support the scalability of the proposed method and address the concerns raised by Reviewer **iFFC** and **C7fR**.

4. **Analysis of interpretablity**

   We validated through experiments how energy estimation and evolutionary prediction influence each other’s performance, which further strengthens the motivation of this paper. This was acknowledged by Reviewer **iFFC**.

5. **Additional ablation study**

   We added ablation experiments on the loss function and key modules, addressing the concerns raised by Reviewer **C7fR** and **jDjA**.

6. **Clarification of experiment protocols**

   Following Reviewer **ciD9**'s suggestion, we clarified the consistency and fairness of the experimental setup and evaluation metrics with respect to prior work. Additionally, we added supplementary experiments on the degradation of the observation function g to ensure the completeness of the experimental setup.

7. **Polished writings**

   We refined the introduction of the proposed method and model components, and added analyses of the model architecture, loss function, and applicable scenarios to make the work more accessible to the audience. This was acknowledged by Reviewers **iFFC**, **jDjA**, and **C7fR**.

All updates are highlighted in red. Compared with the first submissions, $\underline{\text{the revised paper}}$ has **additional 7 pages**.

We sincerely appreciate the active and constructive feedback from the reviewers. All reviewers who have participated in the discussion so far have acknowledged that we have addressed their concerns and provided positive rating.  Although Reviewer **ciD9** has not provided further feedback or participated in the discussion, our responses to similar concerns (generalization and method motivation) have been acknowledged by the other reviewers (**C7fR** and **iFFC**). Additionally, we have provided the additional experiments and clarifications in response to his questions regarding the experimental setup. We believe that the concerns have been thoroughly addressed in our responses and revisions.

The valuable suggestions from reviewers are very helpful for us to revise the paper to a better shape. We'd be very happy to answer any further questions.

---

### Meta-Review · Area_Chair_5Amn · 2024-12-23

**Metareview:**

This paper is on estimating the gradient dynamics of a scalar energy field. For dimensionality reduction, vector quantization techniques are proposed wherein the latent space is discretized, and the energy field is approximated using the energy of a codeword. To respect locality of codeword regions and model the time-dependent evolution of the probability distribution over codewords, a graph-based Fokker-Planck equation is formulated. Experiments are reported on three problems: 2d Prinz potential, ecological evolution, and fast-folding peptides.

Strengths: Good combination of ideas: vector-quantization, self-supervised formulation etc.  Encouraging experimental results.

Weaknesses:  Presentation clarity is inconsistent.  Generalizability of the approach outside of visited regions. Limited ablations and lack of comparisons with published results.

**Additional Comments On Reviewer Discussion:**

The authors had strong engagement during the rebuttal process and significantly enhanced the paper with improved presentation clarity, more empirical analysis and ablations on the effect of loss term; more explanation of Graph Neural FP equation, etc. The overall reviews are favorable for acceptance.

---

### Decision · Program_Chairs · 2025-01-22

Accept (Poster)